# Voltage gating of mechanosensitive PIEZO channels

Mirko Moroni[1], M. Rocio Servin-Vences[1], Raluca Fleischer[1], Oscar Sánchez-Carranza[1] & Gary R. Lewin [1,2]

Mechanosensitive PIEZO ion channels are evolutionarily conserved proteins whose presence is critical for normal physiology in multicellular organisms. Here we show that, in addition to mechanical stimuli, PIEZO channels are also powerfully modulated by voltage and can even switch to a purely voltage-gated mode. Mutations that cause human diseases, such as xerocytosis, profoundly shift voltage sensitivity of PIEZO1 channels toward the resting membrane potential and strongly promote voltage gating. Voltage modulation may be explained by the presence of an inactivation gate in the pore, the opening of which is promoted by outward permeation. Older invertebrate (fly) and vertebrate (fish) PIEZO proteins are also voltage sensitive, but voltage gating is a much more prominent feature of these older channels. We propose that the voltage sensitivity of PIEZO channels is a deep property co-opted to add a regulatory mechanism for PIEZO activation in widely different cellular contexts.

[1] Department of Neuroscience, Max-Delbrück Center for Molecular Medicine, Robert-Rössle Straße 10, D-13092 Berlin, Germany. [2] Excellence Cluster Neurocure, Charité Universitätsmedizin, 10117 Berlin, Germany. Correspondence and requests for materials should be addressed to M.M. (email: mirko.moroni@mdc-berlin.de) or to G.R.L. (email: glewin@mdc-berlin.de)

The mechanically gated ion channels PIEZO1 and PIEZO2 are involved in a variety of physiological functions essential to life. PIEZO1 plays a fundamental role in the development of the mouse vasculature as well as lymphatic systems[1–3]. The PIEZO2 protein is found in sensory neurons of the dorsal root ganglia and Merkel cells where its presence is essential for mechanotransduction[4–6]. Mice without *Piezo2* in sensory neurons lack normal touch sensation and proprioception, and severe loss of function alleles in humans are also associated with loss of proprioception and touch sensation[4,7–9]. Genetic ablation of either the *Piezo1* or *Piezo2* genes in mice leads to either embryonic or early post-natal lethality[1,2,10]. Recently, both mouse and human genetics have revealed roles for PIEZO-mechanosensing ion channels in a variety of non-sensory cellular physiology ranging from cartilage formation by chondrocytes[11,12], epithelial sheet homeostasis[13,14], growth cone guidance[15], arterial smooth muscle remodeling[16] to blood cell shape regulation[17].

The mouse PIEZO1 protein forms a pore-forming ion channel directly gated by membrane stretch[18–20]. High resolution structures of the mouse PIEZO1 protein were recently obtained revealing a trimeric three-bladed, propeller-shaped structure with a central pore-forming module comprising of an outer helix (OH), C-terminal extracellular domain (CED), inner helix (IH), and intracellular C-terminal domain (CTD)[21–23]. The peripheral regions of PIEZO1 are composed of the extracellular "blade" domains, 36 peripheral helices (PHs) in each subunit and intracellular "beam" and "anchor" domains[23]. The PIEZO1 structure has facilitated biophysical exploration of the ion-channel pore[24–28], and chimeric structures suggest that the N-terminal non-pore containing region confers mechanosensitivity on the channel[28]. However, recent data suggested that these models need further validation[29].

Proteins like STOML3 have been identified that can dramatically increase the sensitivity of PIEZO channels to mechanical force[30,31]. However, the role of membrane voltage in modulating PIEZO channel activity has barely been addressed despite the fact that both mammalian stretch-activated potassium channels[32–36], as well as bacterial stretch-activated channels are voltage sensitive[37]. Here, we asked whether voltage can modulate or even gate vertebrate and invertebrate PIEZO channels. We show that both PIEZO1 and PIEZO2 show significant voltage sensitivity that is dependent on critical residues in the pore-lining region of the channel. We provide evidence for an inactivation gate that closes following inward permeation which under physiological conditions renders >90% of the channels unavailable mechanical gating. Outward permeation of the channel is sufficient to initiate a slow conformational change that opens the inactivation gate. Pathological human mutations in *Piezo1* primarily weaken the inactivation gate and render the channels less sensitive to voltage modulation. The same mutations also allow PIEZO1 to behave as a voltage-gated ion channel in the absence of mechanical stimuli. Finally, we show that the biophysical properties of both invertebrate (*Drosophila melanogaster*) and other vertebrate (*Danio rerio*) PIEZOs are much more reminiscent of classical voltage-gated channels than mammalian PIEZOs.

## Results

### PIEZO1 open probability is increased at positive voltages.

PIEZO1 channels show a linear I/V relationship between −50 and +50 mV, but the current inactivation is much slower at positive holding potentials compared to negative voltages[5,28]. We subjected excised outside-out patches from the N2a cells of the overexpressing mouse PIEZO1 to an I/V protocol from −100 to +100 mV in symmetrical Na⁺, without divalent cations, while stimulating the patch with a saturating pressure pulse. However, at more positive voltages the I/V relationship showed strong outward rectification (rectification index $I_{-60\,mV}/I_{60\,mV}$ 5.3 ± 0.6, 12 cells) (Fig. 1a). Single PIEZO1 channels may conduct more ions outward than inward, a phenomenon seen in glycine and GABA$_A$ channels[38], however, single channel measurements under identical conditions showed that channel outward conductance was smaller than the inward conductance (Fig. 1b).

If outward rectification were due to intrinsic pore properties, then measurements of instantaneous currents to voltage steps should show the same rectification index. We took advantage of the slow inactivation kinetics of PIEZO1 at positive voltages to trigger the channel opening at +60 mV with a saturating pressure pulse (70 mmHg). Upon reaching steady-state activation, the voltage was stepped to −60 mV and the peak instantaneous currents were measured. The rectification index ($I_{ins-60\,mV}/I_{60\,mV}$) was 1.13 ± 0.06 ($n = 8$) (Fig. 1c), suggesting that the pore conducts macroscopic current approximately equally in both directions. We speculated that not only pressure, but also voltage could affect the apparent open probability of PIEZO1 and tested this idea by measuring tail currents. A saturating pressure pulse was applied simultaneously with voltage steps ranging from 0 to +150 mV (Fig. 1d). Upon reaching steady-state activation, the voltage was stepped to −60 mV in the continuous presence of pressure. We found that the amplitude of inward (tail) currents upon repolarization to −60 mV increased proportionally to the magnitude of the positive pre-pulse voltage, indicating a voltage-dependent increase in maximum apparent open probability. A Boltzmann fit of these data (Fig. 1e) established that voltage is a major contributor to the gating of PIEZO1 and that depolarized voltages increase the open probability ($V_{50} = 91.9 ± 3.2$ mV, slope 22.2 ± 0.9, 12 cells). We have observed a high variation in saturation levels of Boltzmann fits depending on the patch. While some patches saturated at 140 mV, others did not. Higher voltages might achieve consistent saturation levels, but were not achievable due to frequent patch rupture. Examination of the Boltzmann fit indicates that at physiological resting membrane potentials (<0 mV), the apparent open probability, or PIEZO1 channel availability was <10% of the maximum made available by a depolarizing pre-pulse (Fig. 1e).

### Outward permeation resets PIEZO1 kinetics.

Mechanical stimulation of PIEZO1 via pressure, cell poking, or substrate deflection at negative potentials show fast current inactivation[5,12,18,28,31], and here we also measured fast inactivation time constants ($\tau_{inact} = 60.9 ± 3$ ms, eight cells). It has been reported that repetitive mechanical stimulation can drive PIEZO1 channels into an irreversible non-inactivating and desensitized state[31,39]. The desensitized state, induced by repetitive stimulation (every 1 s), is characterized by a loss of inactivation as reflected in a reduction in the ratio of peak to steady-state current (Fig. 2a). Interestingly, mechanical stimuli applied every second at positive voltages (+60 mV) induced no loss of PIEZO1 inactivation and only minimal desensitization (Fig. 2b, d). PIEZO1 is involved in a variety of biological processes where pressure, stretch, or shear stress are constantly monitored[1,2,16] thus there might be a mechanism to release PIEZO1 channels from a desensitized and non-inactivating state. By analogy to fast inactivating voltage-gated sodium channels where hyperpolarization allows channels to rapidly exit inactivation, we asked whether positive voltages could reset PIEZO1 channels by applying constant pressure pulses at alternating negative and positive voltages (Fig. 2c). Using this protocol, desensitization of PIEZO1 currents induced by repetitive stimuli was completely abolished (Fig. 2c, d). Positive voltages also prevented the channel from acquiring non-

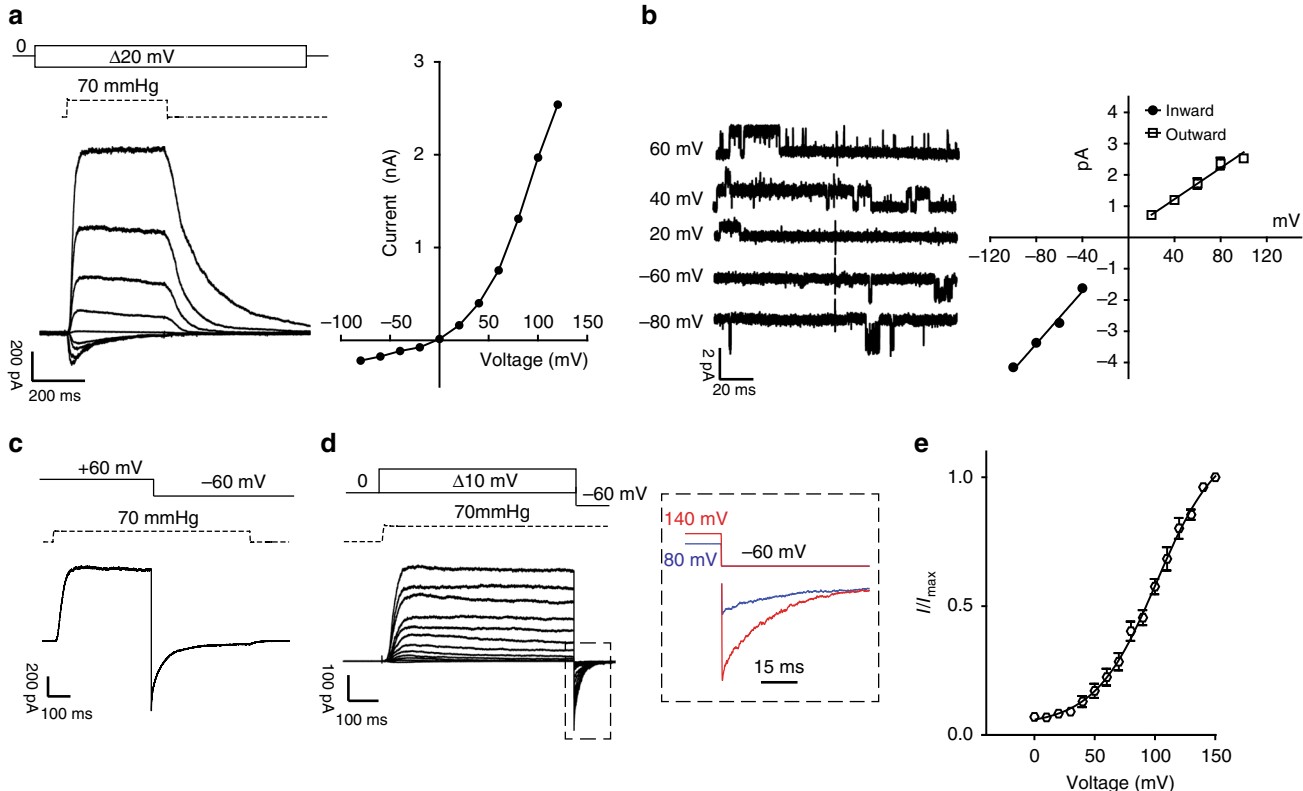

**Fig. 1** Rectification and voltage modulation of pressure-mediated PIEZO1 currents. **a** Left: example traces of currents elicited at a constant saturating pressure (70 mmHg) and at increasing voltages (in 20 mV steps from −100 to 100 mV) in symmetrical Na$^+$ from excised outside-out patches overexpressing mPIEZO1 in N2a cells. Right: peak currents are plotted against voltage to show an I/V relationship. Note the outward rectifying behavior. **b** Left: single channel openings were recorded at negative and positive voltages to obtain the slope-conductance values. Right: linear regressions from individual patches were averaged and pooled. Inward slope conductance was significantly higher than outward slope conductance (41.3 ± 0.9 pS and 27.1 ± 1.2 pS, respectively, three cells. Student's t-test $P = 0.0006$, $dF = 4$). **c** Example trace of instantaneous currents recorded upon switching voltage from +60 to −60 mV during the application of a 70 mmHg pressure stimulus (rectification index 1.13 ± 0.06 ($n = 8$)). Capacitance currents were digitally subtracted. **d** Current responses to a pressure stimulation of 70 mmHg during 300 ms voltage steps ranging from 0 to 150 mV, followed by a repolarization step to −60 mV to obtain tail currents. The inset shows an expanded example of tail currents at −60 mV originated from a pre-stimulation step at 80 (blue) and 140 mV (red) in the presence of 70 mmHg of pressure. **e** Tail currents from individual cells were normalized to their maximum and fitted to Boltzmann relationship ($V_{50} = 91.9 ± 3.2$ mV, slope 22.2 ± 0.9, 12 cells). Pooled data are shown as mean ± SEM

inactivating kinetics (Fig. 2c). Since different channel-states likely correspond to distinct structural conformations, we assume that positive voltage together with outward permeation may hold the channel in a fully active conformation preventing the desensitized state.

We next asked whether outward currents could restore an active conformational state after channel desensitization. We applied three pressure pulses at negative voltages followed by one at a positive voltage and repeated the sequence without a pause (Fig. 2e). After three pressure pulses (P1–P3), PIEZO1 currents showed marked desensitization (Fig. 2e). Once outward permeation of PIEZO1 occurs by stepping to +40 mV, the current amplitude of pulse 5 (P5) measured at −60 mV was almost identical in amplitude and kinetics to P1 (Fig. 2e, g) (P1 $\tau_{inact}$ 62.3 ± 2.1 ms, P5 61.4 ± 3.1 ms eight cells). This experiment demonstrates how voltage or outward permeation can reset the channel to a fully active state. To address whether permeation or positive voltage were necessary to reset desensitization, we repeated the experiments as above but did not apply pressure pulse P4 while holding the patch at +40 mV. Under these conditions, no outward current was measured and PIEZO1 channels did not recover from desensitization; the P5/P1 was decreased to 0.25 ± 0.05 ($n = 13$) (Fig. 2f, h).

We altered the reversal potential by lowering the intracellular Na$^+$ concentration (NaCl 10 mM inside, NaCl 140 mM outside) and measured the $E_{rev}$ of +43 ± 1 mV (five cells) with a ramp protocol (Supplementary Fig. 1a), a similar $E_{rev}$ of +45 ± 2 mV, three cells were measured using a voltage step protocol (Supplementary Fig. 1a, b). Using these ionic conditions, we repeated the experiment in Fig. 2e, and as expected obtained inward currents at P4 at the +40 mV holding potential (Fig. 2g, i). Under these conditions the ratio between P5/P1 was 0.28 ± 0.06 ($n = 7$) (Fig. 2h). These experiments established that the loss of inactivation is not an irreversible event, as has previously been suggested[40], and that outward-ion conduction and not positive voltage per se is sufficient to reset PIEZO1 channels.

We noticed that the transition between the desensitized state (after three pressure pulses at −60 mV, P1–P3 Fig. 2e) and a "recovered" active state (P5) may require a slow conformational change as reflected in the slow rise in current amplitude during P4. To explore this idea further, we made use of a paired-pulse protocol and compared the peak currents elicited by the pressure pulses at +60 mV preceded by a pressure pulse at −60 mV (blue) or +60 mV (black) (Fig. 2j). Currents evoked by the second pressure pulse showed a threefold slower rise time and 25% reduced amplitude when preceding by a pressure pulse at −60

mV compared to +60 mV (Fig. 2j, k). Increasing the interval by up to 30 s between the two pulses did not alter this effect (Fig. 2l).

In order to assess the rate of exit from inactivation, we modified our paired-pulse protocol by adding a pressure step at positive voltages between the pulse at −60 and +60 mV. We first varied the voltage (from +10 to +80 mV) during the second pressure step, and measured the changes in amplitude and rise time of the last step at +60 mV (Supplementary Fig. 1c). The

current rise time decreased and amplitude increased proportional to the voltage applied in the conditioning step. Using a second protocol with conditioning pressure pulses applied at +60 mV (from 20 to 160 ms), we found that the rise time also decreased and amplitude increased with longer pressure steps (Supplementary Fig. 1d). Thus both the duration and amplitude of the conditioning step determine the rate at which PIEZO1 is released from its inactive state.

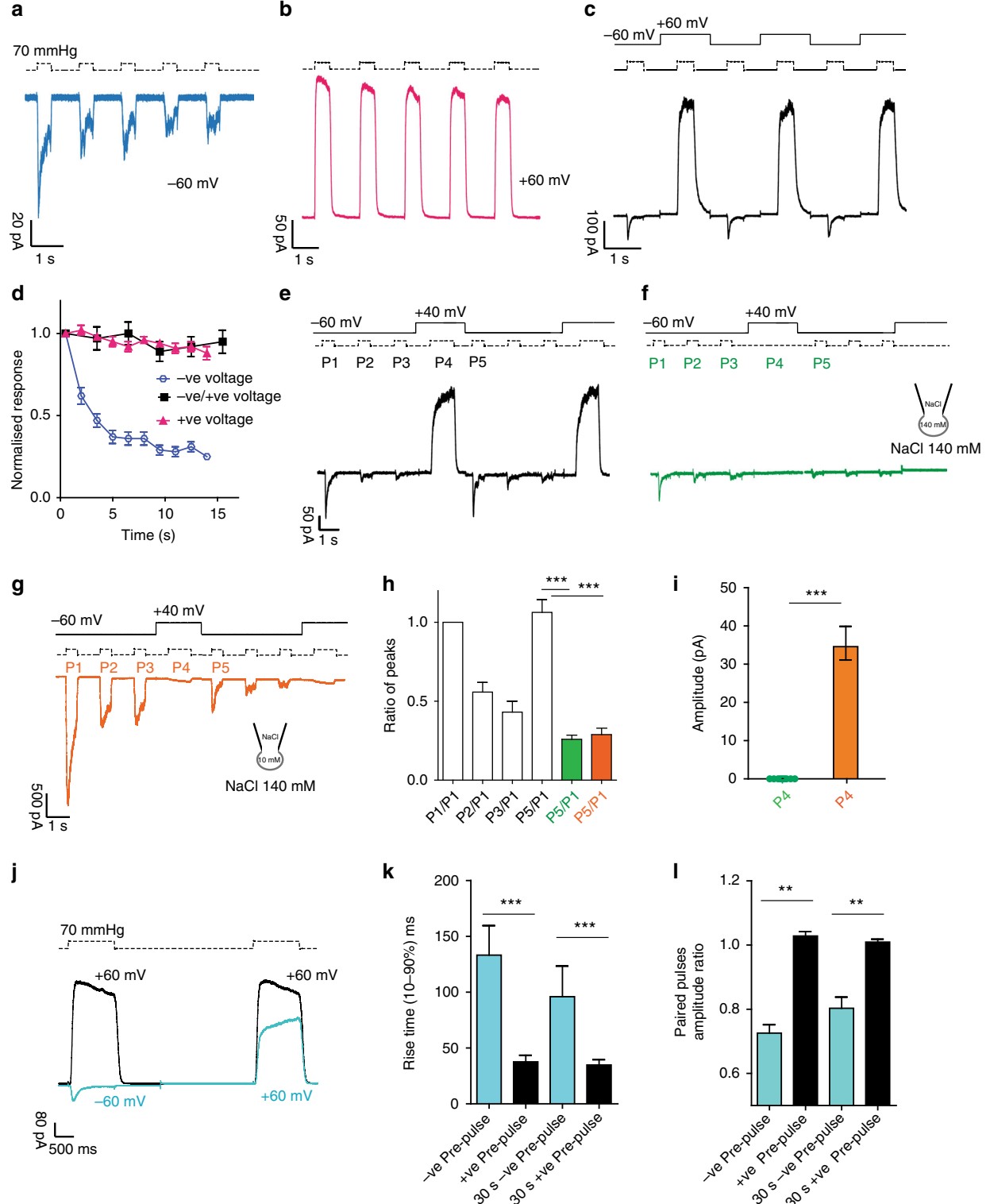

**Removal of inactivation requires outward permeation.** Since outward permeation resets channel properties, we hypothesized that outward-ion flux should remove inactivation in a manner proportional to the driving force. We recorded currents elicited by pressure pulses at $-60$ mV that were preceded by conditioning pressure steps with increasing driving force (voltages from $+20$ to $+140$ mV). We found that larger outward currents measured in the conditioning step were followed by larger inward currents measured at $-60$ mV (Fig. 3a, b). By plotting the conditioning voltage vs the peak current amplitude at $-60$ mV, we calculated that removal of inactivation had a $V_{50}$ of $85.5 \pm 5$ mV ($n = 10$). Thus, the inactivation state of PIEZO1 was reversible and the transition occurs in a manner proportional to the amount of permeating ions. There is thus a mechanistic link between permeation and inactivation which implies the presence of an inactivation gate along the permeation path. The opening of an inactivation gate would depend on the outward flow of ions through the pore. Furthermore, this inactivation gate can be forced open by increasing the outward driving force during permeation, thus establishing an essential role for permeating ions in the gating of PIEZO1 (Fig. 3e).

We added a conditioning pressure pulse at $+100$ mV to the tail current protocol. The conditioning pressure pulse at positive potentials removes ~80% of the inactivation (Fig. 3a, b), and by plotting tail current amplitudes against voltage we measured a substantial (~20 mV) leftward shift in the current–voltage relationship ($90.6 \pm 2.9$ mV, $n = 12$ without conditioning step, black trace, and $68.3 \pm 3.7$ mV $n = 5$, with conditioning step, red trace, Student's $t$-test $p = 0.0036$, dF $= 15$). Thus, outward permeation increases the apparent open probability of the channels. The shift in $I/I_{max}$ relation induced by a positive pre-pulse effectively leads to a threefold increase in the number of available channels at physiological membrane potentials ($<0$ mV) from 5 to 15% (Fig. 3d).

**Altered voltage modulation of PIEZO1 in human disease.** Missense mutations in the coding sequence of Piezo1 lead to diseases, such as xerocytosis and lymphatic dysplasia[3,17,41–45]. Xerocytosis mutations slow channel inactivation to membrane stretch or cell poking[3,17,42]. We introduced the missense mutation R to H at position 2482 in the mouse cDNA (human R2456H), a residue located at the base of the pore-forming IH in the CTD of mPIEZO1[21,23,28]. The human R2456H mutation has a strong inactivation phenotype[17,42,46], which we confirmed for the mouse PIEZO1 R2482H mutant (Supplementary Fig. 2a). The desensitization properties of the mouse PIEZO1 R2482H mutant upon repetitive mechanical stimuli at negative potentials ($-60$ mV) were indistinguishable from the wild type (Supplementary Fig. 2b). Current amplitude was also maintained when stimulating at positive holding potentials ($+60$ mV) and alternating positive/negative voltages ($-60/+60$ mV) as for the wild-type channel (Supplementary Fig. 2c, d). The slow transition out of the inactive/desensitized state (measured from the rise time of a pressure pulse at $+60$ mV after three pulses at $-60$ mV) was also similar to wild type (Supplementary Fig. 2e,f).

We measured the apparent open probability with tail current protocols and observed a 50 mV leftward shift in the current–voltage relationship for the R2482H mutant compared to its wild type. When R2482 was mutated to lysine (K)[46], the current–voltage relationship was shifted even further to negative voltages (R2482H: $V_{50}$ $42.0 \pm 3.7$ mV, 19 cells, R2482K: $V_{50}$ $38.0 \pm 2.7$ mV, 6 cells, both significantly different from the wild type $V_{50}$ $91.8 \pm 3.2$ mV, 7 cells, $p < 0.0001$, one-way ANOVA and Dunnett's post-hoc test) (Fig. 4). Interestingly, both the R2482H and R2482K mutants showed a much higher apparent open probability at negative voltages than the wild-type channels (compare tail currents in Fig. 4a and current–voltage relationship in Fig. 4b). The neutralization of the Arg residue to Ala also produced a similar effect to R2482H (Fig. 4b), suggesting that the charged His might not be critical for the human phenotype (R2482A $V_{50}$ $45.5 \pm 5.4$ mV, 10 cells). We introduced further xerocytosis mutations into the mouse cDNA: R1353P (human R1358P[17]) is located on the beam structure and may interact with the CTD[23], A2036T (human A2020T[17]) is located on a peripheral membrane helix that is resolved in the structure, and T2143M (human T2127M) is located in the middle of the anchor domain (Fig. 4c). In contrast to the R2482H and R2482K mutants, the peripherally located A2036T and R1353P mutations had no effect on the voltage sensitivity of the channel (Fig. 4a, b). However, the A2036T and R1353P mutations did significantly slow the inactivation kinetics of the channel (Supplementary Table 1)[17]. In contrast, the T2143M anchor site mutation had a very similar effect on the R2482H mutation in altering the voltage sensitivity of the channel. The currently available structures provide atomic resolution only for the extracellular CED region, and much less information of the PHs[21–23]. However, structurally the mutated threonine (2143) is predicted to be located about 17 Å distant from arginine (2482) that is located on the adjacent subunit. Thus, both residues R2482 and T2143 could participate in a spatially restricted domain that controls voltage-dependent gating of PIEZO1 channels.

**Voltage modulation and stretch activation of PIEZO chimeras.** PIEZO2 has a fundamental role in sensory neuron mechanotransduction where it is necessary for mechanoreceptor function and touch[4,5,8,9,47]. Mechanically activated PIEZO2 currents inactivate faster than PIEZO1 currents[5,31]. We hypothesized that the apparent open probability of PIEZO2 channels are also

**Fig. 2** Inactivation and desensitization of PIEZO1 can be reset by outward permeation. Repetitive pressure stimulations desensitize PIEZO1 at $-60$ mV (**a**) but not at $+60$ mV (**b**). Note the decreased peak current and low peak/steady-state current ratio at $-60$ mV. **c** Alternating pressure pulses at $-60$ mV and $+60$ mV abolishes desensitization and prevents PIEZO1 from entering a non-inactivating state. **d** Peak currents in **a** (negative pulses), **b** (positive pulses) and **c** (negative pulses) were normalized to the first response and plotted against time. Desensitization is absent at alternating voltages. **e** Three pressure pulses (P1, P2, P3) at $-60$ mV were applied to patches expressing PIEZO1 to drive the channel into a desensitized state, followed by one positive pressure pulse (P4) at $+40$ mV. The sequence was repeated twice. Outward permeation (P4) recovers PIEZO1 initial current (P1), as shown by the ratio of P5/P1, shown in **h**). **f** The stimulation sequence in **e** was repeated in absence of pressure at P4. The recovery from desensitization does not occur at $+40$ mV in absence of permeation. **g** The same protocol in Fig. 2e was repeated with the ionic concentrations shown. Inward currents flow at P4 at $+40$ mV. No recovery from desensitization is observed at P5, underlining the importance of outward permeation for resetting channel kinetics. **h** The P5/P1 ratio for **e** (white, $n = 10$), **f** (green, $n = 13$), and **g** (orange, $n = 7$) are shown and are statistically different (Anova, Dunnett's post-hoc test, dF $= 18$, white vs green $P = 0.00004$, white vs orange $P = 0.001$). **i** Raw amplitude levels for P4 in **f** (green) or **g** (orange) are statistically different (Student's $t$-test $P = 0.00003$, dF $= 20$). **j** Paired pulses protocol of a pressure stimulus at $+60$ mV preceded by either a pressure step at $-60$ mV (light blue) or at $+60$ mV (black). The direction of the first current stimulus affects the amplitude and the time course of activation of an identical second step at $+60$ mV. **k** The rise time and the amplitude ratio of the second stimulus at $+60$ mV in **j** are compared. The interval between the two stimuli was either 2 or 30 s. Both parameters are significantly different (Students $t$-test $P < 0.001$, $n = 28$, dF $= 26$). The data are shown as mean $\pm$ SEM

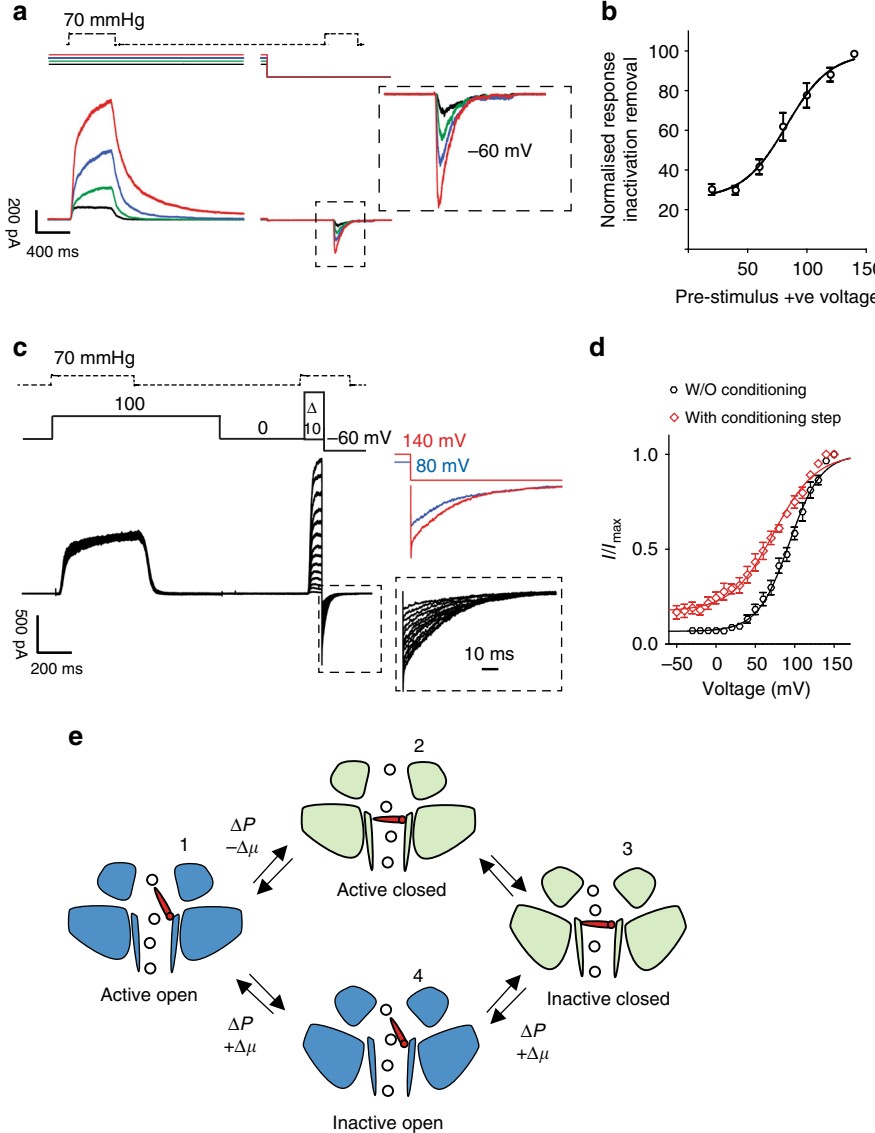

**Fig. 3** Increasing outward permeation determines the number of channels available for activation. **a** A constant pressure pulse at −60 mV was preceded by a constant pressure stimulation (70 mmHg) at increasing voltages (20 mV black, 40 mV green, 60 mV blue, and 80 mV red). The current amplitude of the second pressure stimulation depends on the driving force applied during the preceding step, showing that the larger the applied driving force the greater the relief from inactivation. **b** The conditioning stimulus voltage was plotted against the normalized current amplitude of the currents recorded at −60 mV. Single cells were fitted individually to a Boltzmann fit. The data shown represent pooled data from 10 cells ($V_{50} = 85.5 \pm 5$ mV). **c** Tail current protocol as in Fig. 1d preceded by a conditioning pressure pulse at +100 mV to remove ~80% of inactivation. **d** Tail currents from individual cells were normalized to their maximum and fitted to a Boltzmann relationship ($V_{50} = 90.6 \pm 2.9$ mv, slope $24.6 \pm 1.6$, $n = 12$ cells without the conditioning step, black trace, $V_{50} = 68.3 \pm 10.2$ mV, slope $43.7 \pm 2.3$, 5 cells, with conditioning step at +100 mV, red trace, unpaired Student's $t$-test $P = 0.0036$, d$F = 15$). Pooled data are shown as mean ± SEM. **e** Proposed transition between active and inactive state PIEZO1. Activation by pressure ($\Delta P$) and negative electrochemical gradient ($-\Delta \mu$) drives the channel into an inactive state (domains tilted upward). Pressure and positive electrochemical gradient ($+\Delta \mu$) reset the channel to an active state by opening an inactivation gate. The transition 3-4-1 is a slow conformational change as suggested by the rise time and amplitude in Fig. 2j, k, l

modulated by voltage. However, we were unable to record the stretch-activated currents in excised or cell-attached patches from cells overexpressing PIEZO2 (Fig. 5a, c), see also ref. [48]. It is possible that PIEZO2 responds poorly to membrane stretch but is activated in a cellular context by virtue of association with other molecules, a phenomenon described for TRPV4[12]. Using pressure to gate channels in excised patches has many experimental advantages, so we decided to generate two stretch-activated chimeric PIEZO1/PIEZO2 proteins (P1/P2 chimera, P1 aa 1-2190, and P2 aa 2472-2822) and the reverse PIEZO2/PIEZO1 (P2/P1 chimera, P2 aa 1-2471, and P1 aa 2188-2547) by fusing the N-terminal sequence of the first protein and including

the so-called anchor region with the remaining downstream sequences of the second protein. The C-terminal sequence of the chimeras included: the last two transmembrane-spanning domains that putatively form the channel pore, the so-called IH and OH, the CED, and CTD (Fig. 5a). To study the chimeric channel under the same experimental conditions, we used a CRISPR/Cas9 approach to generate a N2a cell line in which the mouse *Piezo1* gene had been deleted (referred to as N2a $^{Piezo1-/-}$ cells, see Methods and Supplementary Fig. 3). We overexpressed both the chimeric channels in N2a $^{Piezo1-/-}$ cells and could record robust currents to soma indentation that had similar amplitudes to wild-type currents (Fig. 5a, b). The inactivation

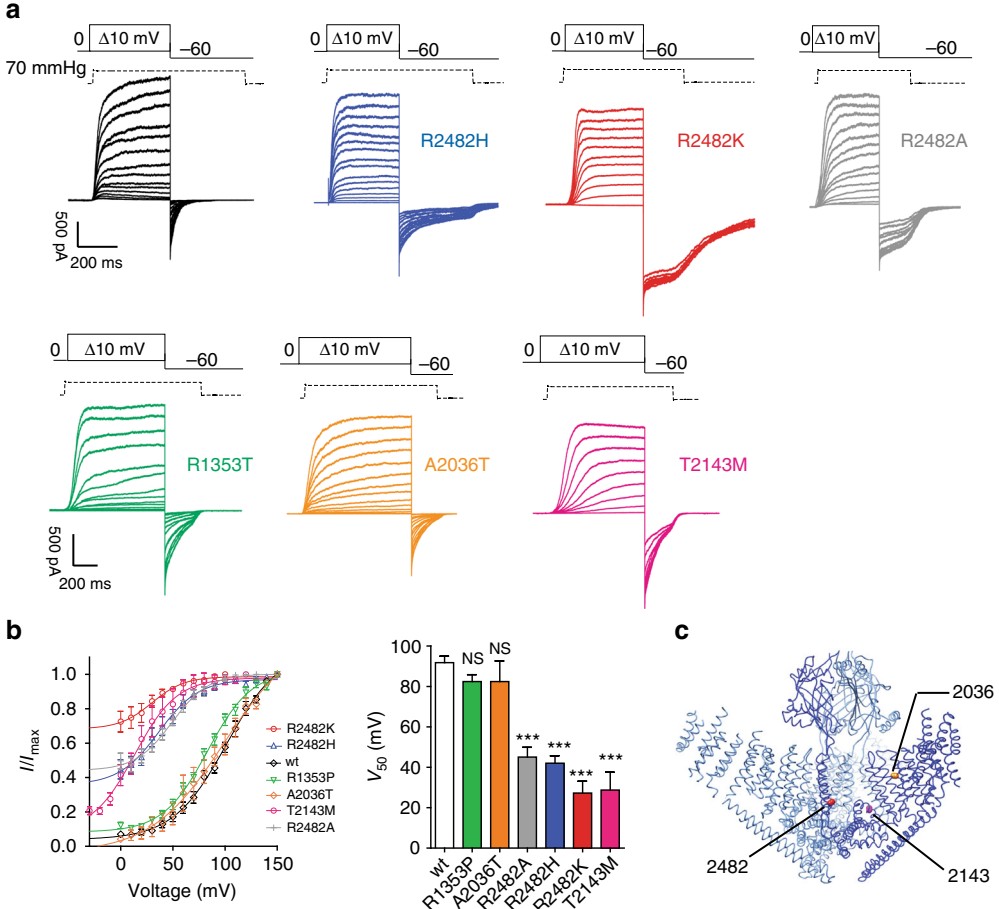

**Fig. 4** Mutations causing xerocytosis in humans alter voltage modulation of PIEZO1. **a** Current responses to a pressure stimulation of 70 mmHg during 300 ms voltage steps ranging from 0 to 150 mV, followed by a repolarization step to −60 mV to obtain tail currents for wild-type PIEZO1 (black), R2482H (blue), R2482K (red), R1353T (green), A2036T (orange), T2143 (magenta), and R2482A (gray) mutants. The R2482K mutant shows decreased voltage modulation. **b** Tail currents from individual cells were normalized to their maximum and fitted to a Boltzmann relationship (wt: $V_{50} = 91.9 \pm 3.2$ mV, slope $22.2 \pm 0.9$, 12 cells; R2482H $V_{50} = 42.0 \pm 3.7$ mV, slope $14.4 \pm 1.5$, 19 cells; R2482K $V_{50}$ $27.2 \pm 5.9$ mV, slope $12.2 \pm 1.5$, 10 cells, R1353P $V_{50} = 82.4 \pm 8.8$ mV, slope $20.1 \pm 1.3$, 8 cells, A2036T $V_{50} = 82.4 \pm 10.2$ mV, slope $22.4 \pm 0.9$, 5 cells; T2143M $V_{50} = 28.7 \pm 8.8$ mV, slope $17.6 \pm 2.0$, 8 cells; R2482A $V_{50} = 45.5 \pm 5.4$ mV, slope $15.1 \pm 2.1$, 10 cells. One-way ANOVA, Dunnett's post-test, significance $P < 0.0001$, $dF = 72$). Pooled data are shown as mean ± SEM. **c** 3D structure of the trimeric mPIEZO1 showing the position of the mutants involved in xerocytosis[21]. R1353 is located on the beam and is not shown. R2482 is in the bottom of the inner helix, A2036 is located in a peripheral helix, and T2143 is in the anchor domain. Molecular graphic was performed with the UCSF Chimera package

kinetics of the poking-induced current reflecting the properties of the protein from which the pore module was derived (Fig. 5a, b), suggesting that the pore module governs inactivation kinetics.

In excised patches from cells expressing the P1/P2 chimeric channel, we recorded robust stretch-activated currents (Fig. 5c). Much lower current amplitudes were observed for the P2/P1 chimera (Fig. 5d). The pressure sensitivity of both P1/P2 and P2/P1 chimeric channels was very similar to that of PIEZO1 channels, $P_{50}$ for mPIEZO1, P1/P2 and P2/P1 were $38.9 \pm 3.1$ mmHg, (21 cells); $37.6 \pm 4.6$ mmHg (10 cells) and $42.4 \pm 4.7$ mmHg, respectively (Fig. 5c, e). The inactivation kinetics of P1/P2 chimeric currents were approximately threefold faster than PIEZO1 ($26.3 \pm 6.2$ ms, 4 cells for P1/P2 channel and $60.9 \pm 3.0$ ms, 9 cells for wt PIEZO1) (Fig. 5b, f) and thereby showed similar fast inactivation kinetics seen for poking-induced PIEZO2 currents. Surprisingly, the P2/P1 chimera did not show any inactivation (Fig. 5c, 10 cells). Considering that the PIEZO2 protein was not activated by membrane stretch, our results indicate that membrane stretch is poorly transmitted by the N-terminal of PIEZO2 to the pore module of PIEZO1. Alternatively,

the pore module of PIEZO1 may be contributed to the stretch sensitivity of the P2/P1 construct.

The chimeric P1/P2 channel also showed apparent rectifying behavior (Supplementary Fig. 4a), and voltage-modulation experiments showed current desensitization at negative voltages, but barely at positive potentials (Supplementary Fig. 4b). As for PIEZO1, the P1/P2 chimera conducted the same amplitude of current when stimulated at alternating negative and positive voltages. We also showed that P1/P2 chimera current amplitudes were restored to control values by outward permeation (Supplementary Fig. 4c).

The chimeric P1/P2 channel also produced robust tail currents at −60 mV (Fig. 5f, g) the amplitude of which increased with more positive pre-pulse voltages, indeed the apparent open probability continued to increase to voltages as high as +150 mV. Similarly to the xerocytosis mutations, the lack of inactivation kinetic in the P2/P1 chimera in outside-out patches produced a channel almost insensitive to the voltage ($V_{50}$ $47.9 \pm 6.3$ mV, slope $25.8 \pm 6.3$, 10 cells). These data show that the pore module of PIEZO2 in combination with the N-terminal region of PIEZO1 is also capable of sensing changes in voltage. However, the

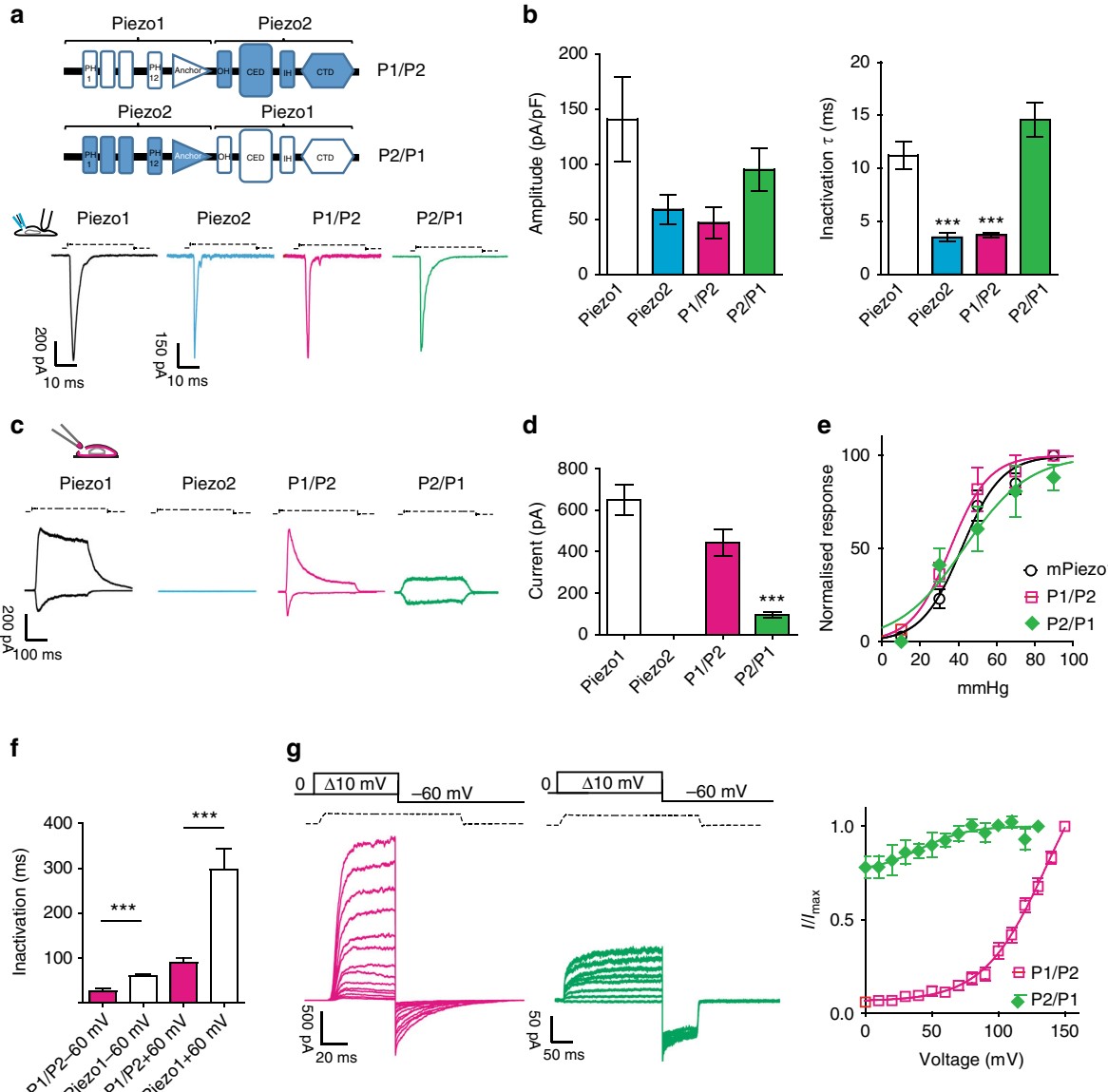

**Fig. 5** PIEZO chimeras respond to membrane stretch. **a** Chimeric PIEZO channels were constructed by fusing the N-terminal region of one protein to the pore domain of the other protein including the CED region. PIEZO1, PIEZO2, and the chimeric channels were overexpressed in N2a $^{Piezo1-/-}$ cells (Supplementary Fig. 3). Cells were clamped at −60 mV and subjected to soma indentation. **b** The maximum current amplitude (normalized for the capacitance) is plotted. The time course of inactivation for each construct was fitted to a mono-exponential function. PIEZO1 inactivation was approximately threefold slower than PIEZO2 and the chimeric receptor was P1/P2 (ANOVA, d$F$ = 25, Dunnett's post-hoc test $P$ < 0.023). The P1/P2 chimera showed a current amplitude and a time course of inactivation similar to PIEZO2. The P2/P1 chimera showed a current amplitude and a time course of inactivation similar to PIEZO1. **c** Typical response of the chimeric channels and wild-type PIEZO1 to membrane stretch in outside-out patches. Outside-out patches pulled from cells overexpressing PIEZO2 did not respond to stretch stimulation. **d** Maximal current level recorded in outside-out patches at −60 mV. Current levels for P2/P1 were significantly different from PIEZO1 patches ($n$ = 10, Student's $t$-test, $P$ = 0.0003, d$F$ = 25. **e** Pressure–response relationships of the chimeric channels are not different from PIEZO1 (P1/P2 37.6 ± 4.6 mmHg, 10 cells; PIEZO1 38.9 ± 3.1 mmHg, 21 cells; P2/P1 42.4 ± 4.7 mmHg, 7 cells, Anova Dunnet's post-test $P$ = 0.78, d$F$ = 32). **f** The decay of inactivation for pressure-mediated responses at +60 and −60 mV is plotted for the chimeric and PIEZO1 channels. Also for pressure-mediated responses, the kinetic of inactivation of PIEZO1 remained approximately threefold slower than the chimeric channel P1/P2 (P1/P2 26.3 ± 6.2 ms, 10 cells; PIEZO1 60.9 ± 5.1 ms at −60 mV, 9 cells, Student's $t$-test, $P$ = 0.0025, d$F$ = 17). The P2/P1 chimera did not show any inactivation properties. **g** Example traces of tail current protocols (at 70 mmHg) for chimeric channels ranging from 0 to 150 mV, followed by a repolarization step to −60 mV. Tail currents (−60 mV) from individual cells were normalized to their maximum and plotted against voltage ($n$ = 10). Pooled data mean ± SEM are shown

unexpected kinetics of the P2/P1 chimeric channel limits conclusions about the contribution of the PIEZO2 N-terminus to voltage modulation.

**Voltage gating of PIEZO1**. The apparent open probability of PIEZO1 and a P1/P2 chimera can be dramatically increased by

depolarized membrane potentials. We hypothesized that a prerequisite for voltage alone to gate PIEZO1 would be a fully open inactivation gate. The R2482H/K mutant channels display a very high apparent open probability and might fulfill such a condition (Fig. 4). We mechanically activated PIEZO1 channels during step depolarizations from 0 to +80 mV, and as expected outward

currents start to decay immediately after cessation of the pressure pulse reflecting deactivation of the channels (Fig. 6 and ref. [5]). However, the R2482H and R2482K mutant channels showed an unexpected additional feature: current deactivation was seen immediately after the pulse, but 20–40 ms later the current started to reactivate with a sag phase and increased or maintained outward current (Fig. 6a). The reactivation of the mutant PIEZO1 outward current was only seen at more depolarized potentials and the mean threshold for this effect was $34.4 \pm 3.4$ mV (9 cells). Thus, voltage might play a direct role in (re-)opening PIEZO1 channels. We hypothesized that positive voltage and outward permeation promotes a new stable conformation of mutant PIEZO1, in which the inactivation gate is fully open allowing direct voltage gating in the absence of externally applied pressure. Thus PIEZO1 sequentially switches between two modes of activation: a pressure-gated and a voltage-gated mode.

To test this hypothesis, we next stimulated patches with a pressure pulse at +80 mV and again we observed a sag followed by current reactivation in the absence of pressure (Fig. 6b). The holding voltage was then stepped to +5 mV for 2 s to avoid inward permeation and to allow the channel to completely deactivate. If the stable conformational change after the sag phase corresponded to a transition between pressure-gated and voltage-gated mode, it should be possible to re-open the channel with a depolarizing voltage step. We then stimulated the patches with a series of depolarizing steps (from 0–60 mV in 10 mV increments) in the absence of an externally applied mechanical stimulus. Strikingly, both the R2482H and R2482K mutant channels were strongly activated by the voltage steps in a manner similar to classical voltage-gated ion channels with a $V_{50}$ of $50.7 \pm 9.3$ (10 cells R2482H) and $19.5 \pm 4.5$ mV (5 cells R2482K), respectively (Fig. 6g). This experiment demonstrates that PIEZO1, carrying a human pathogenic mutation, can undergo transition from a pressure-gated mode to a voltage-gated mode provided that the inactivation gate is kept open. If we allowed inward-ion permeation by stepping to negative potentials, there will be no voltage gating (Supplementary Fig. 5e). Voltage-gated currents displayed a sigmoidal time course for activation and voltage dependency with a marked delay, especially with smaller voltage steps (Fig. 6b, c). Thus, PIEZO channels can be gated by either pressure or voltage. Voltage and pressure likely trigger channel opening via two different mechanisms as revealed by extrapolating the gating charges from the Boltzmann fits of the macroscopic conductance (Supplementary Table 2). While in pressure-gated mode the equivalent gating charge for both R2482K and R2482H is approximately 2.2 $e_0$, in voltage-gated mode the value reaches 6 $e_0$. This difference shows that in a voltage-gating mode, a much higher number of charges must translocate across the membrane to drive channel opening.

Next, we asked whether wild-type PIEZO1 channels can also undergo transition from a pressure to voltage-gated mode. We found that less than 5% of the patches expressing the wild-type PIEZO1 exhibited a reactivation that required high pressure and voltages as high as 180 mV (Supplementary Fig. 5c). These extreme stimulation protocols often resulted in patch rupture. We thus used a small molecule modulator of PIEZO1 activity, Yoda1, to promote conformational states of the PIEZO1 that may enter a voltage-gated mode. Yoda1 has been described as an opener of PIEZO1 channels in artificial lipid bilayers, but fails to directly gate the channel in whole-cell configuration and is a poor partial agonist in cell-attached patches[49]. In outside-out patches concentrations of Yoda1 close to the solubility limit of the compound (10 μM), produced no appreciable channel opening when Yoda1 was applied either in the extracellular buffer or via the recording pipette (Supplementary Fig. 5a). However, Yoda1 acted as a gating modulator of PIEZO1 (Supplementary Fig. 5a,b)

by slowing down both the current rise time, inactivation and deactivation time constants[49]. In the presence of 5 μM, Yoda1 PIEZO1 currents showed a sag and a reactivation process starting at positive voltages (mean threshold was $43.3 \pm 5.6$ mV, 6 cells). We then performed a voltage protocol in absence of pressure and we observed robust voltage-activated outward currents with $V_{50}$ of $31.9 \pm 7.4$ mV (5 cells). The speed of current activation increased exponentially with voltage (Fig. 6f) and an estimation of the gating charges from the steepness of the Boltzmann (Fig. 6g) fit yielded a value of $5.3 \pm 1.7$ $e_0$ (Supplementary Table 2).

We established that mammalian PIEZO channels can switch from mechanically gating mode to a voltage-gated mode and that this transition is promoted by mutations near the pore region or by a synthetic gating modulator.

**Evolutionary conservation of PIEZO voltage gating.** The *Piezo1* gene can be found in plants, invertebrates and protozoa[5]. In insects, a role for *D. melanogaster* PIEZO has been established in mechanical nociception[50]. Additionally, the *Piezo1* gene in zebrafish (*D. rerio*) was found to be crucial for epithelial homeostasis[13,14]. Here, we cloned cDNAs encoding the *Drosophila* (DmPIEZO, accession number JQ425255), Zebrafish Piezo1 (DrPIEZO1), and the human PIEZO1 (hPIEZO1) in order to compare the voltage-dependent behavior of PIEZO channels across three orders of the animal kingdom. To study the orthologues under the same controlled experiments, we used the N2a$^{Piezo1-/-}$ cells (see Methods and Supplementary Fig. 3). While the biophysical and mechanosensitive properties of the fly and human PIEZO1 have been extensively characterized[28,50], this is the first biophysical study of DrPIEZO1. DrPIEZO1 showed mechanosensitivity in outside-out patches with a pressure response behavior of $P_{50}$ of $51.0 \pm 2.7$ mmHg ($n = 9$), which is typical for several PIEZO1 channels (Supplementary Fig. 6a). However, the inactivation time constant $\tau$ at $-60$ mV and at saturating pressure was approximately four times slower than the mouse and human orthologues. Interestingly, in contrast to the rectifying behavior of mPIEZO1 (Fig. 1a), the I/V relationship of DrPIEZO1 showed it behaves as a perfectly ohmic channel at voltages between $-100$ and $100$ mV (Supplementary Fig. 6b). The data suggests that there are profound structural differences in the ion permeation pathway between the DrPIEZO1 and the mPIEZO1 (Supplementary Fig. 6c).

The human channel (hPIEZO1) also desensitized quickly in response to consecutive pressure stimuli (Fig. 7a). In contrast, the amplitude of DmPIEZO and DrPIEZO1 currents only attenuated modestly with repeated pressure stimuli (Fig. 7a). We next investigated the voltage-modulation of DmPIEZO and DrPIEZO1 using tail current protocols (Fig. 1d). The apparent open probability of hPIEZO1 at <0 mV was very small (<5%) and thus almost identical to mPIEZO1 (Fig. 7a, b). However, the apparent open probability from a pre-pulse holding potential of 0 mV differed dramatically between hPIEZO1, DmPIEZO, and DrPIEZO1 (Fig. 7d). The apparent open probability of DmPIEZO was only weakly modulated by voltage and we calculated that ~50% of the channels were available from a pre-pulse holding voltage negative to 0 mV (Fig. 7c, d). In contrast, the gating of DrPIEZO1 was unaffected by voltage so that 100% of the channels were available regardless of pre-pulse voltage. In this respect, the virtual absence of voltage-modulation of the DrPIEZO1 protein was very similar to R2482K mPIEZO1 mutants (compare Figs. 7d and 4b). Considering the similarities in voltage-modulation between the mouse PIEZO1 mutant R2482K and the fly and zebrafish PIEZO orthologues, we asked if DmPIEZO and DrPIEZO1 could be directly opened by voltage.

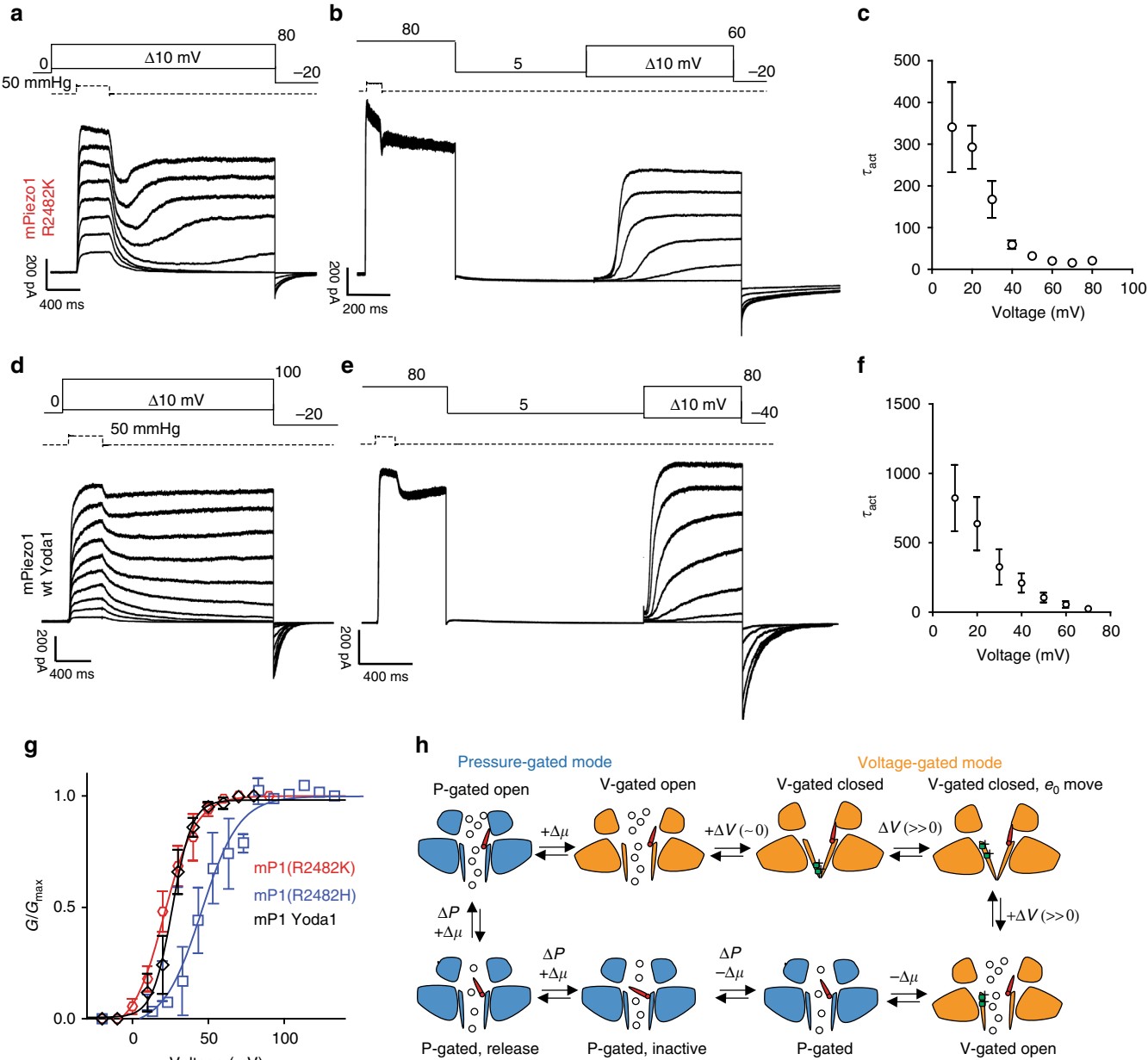

**Fig. 6** PIEZO1 transitions between a pressure-gated and a voltage-gated mode. **a** PIEZO1 R2482K mutant was subjected to pressure stimulations at increasing voltages (from 10 to 80 mV) with a deactivation of 2 s. After an initial deactivation, PIEZO1 at voltages >40 mV undergoes a reactivation phase. **b** A saturating pressure pulse at 80 mV was applied to force the channel to reactivate and switch to a voltage-gated mode. Following a 2 s deactivation period at 5 mV (to prevent the inactivation gate from closing), a family of 1 s steps at increasing voltages (in absence of pressure) was applied, showing that PIEZO1 R2482H/K can be activated by voltage. **c** The mean time, constant of activation (±SEM) for mutant R2482K, is plotted against increasing voltages (n = 10). **d** wild-type PIEZO1 undergoes reactivation in the presence of the gating modifier Yoda1 (5 μM) in the intracellular solution. The same voltage/pressure protocol was applied as in **a**. **e** PIEZO1 can be activated by voltage in presence of Yoda1 (5 μM) intracellularly in absence of externally applied pressure. The same protocol was applied as in **b**. **f** The kinetic of activation decreased at more depolarized voltages as it occurs in voltage-gated ion channels (n = 5). **g** Conductance–voltage relationships for PIEZO1 R2482K (red, n = 10), R2482H (blue, n = 8), wt +Yoda1 5 μM (black, n = 5) in voltage-gated mode were fitted to a Boltzmann equation. The data are displayed as mean ± SEM. **h** Proposed model for gating transitions of PIEZO1. Bottom left: PIEZO1 inactivation gate opens during outward permeation ($+\Delta\mu$) and application of pressure ($\Delta P$) (red inactivation gate partially tilted upward). A persistent depolarization can overcome inactivation and induce a reactivation of the channel (inactivation gate completely open) and a switch to a voltage-gated mode (orange). Deactivation of the channel at voltages close to 0 leads to channel closure. Further depolarization causes a movement of gating charges and opening of the channel. Inward permeation ($-\Delta\mu$) brings the inactivation gate back into the pore and allows the channel to switch back to a pressure-gated mode. Further, pressure-mediated inward permeation leads the channel into an inactive state (inactivation gate tilted toward the center of the pore). Such transition is reversible and mediated further by outward permeation

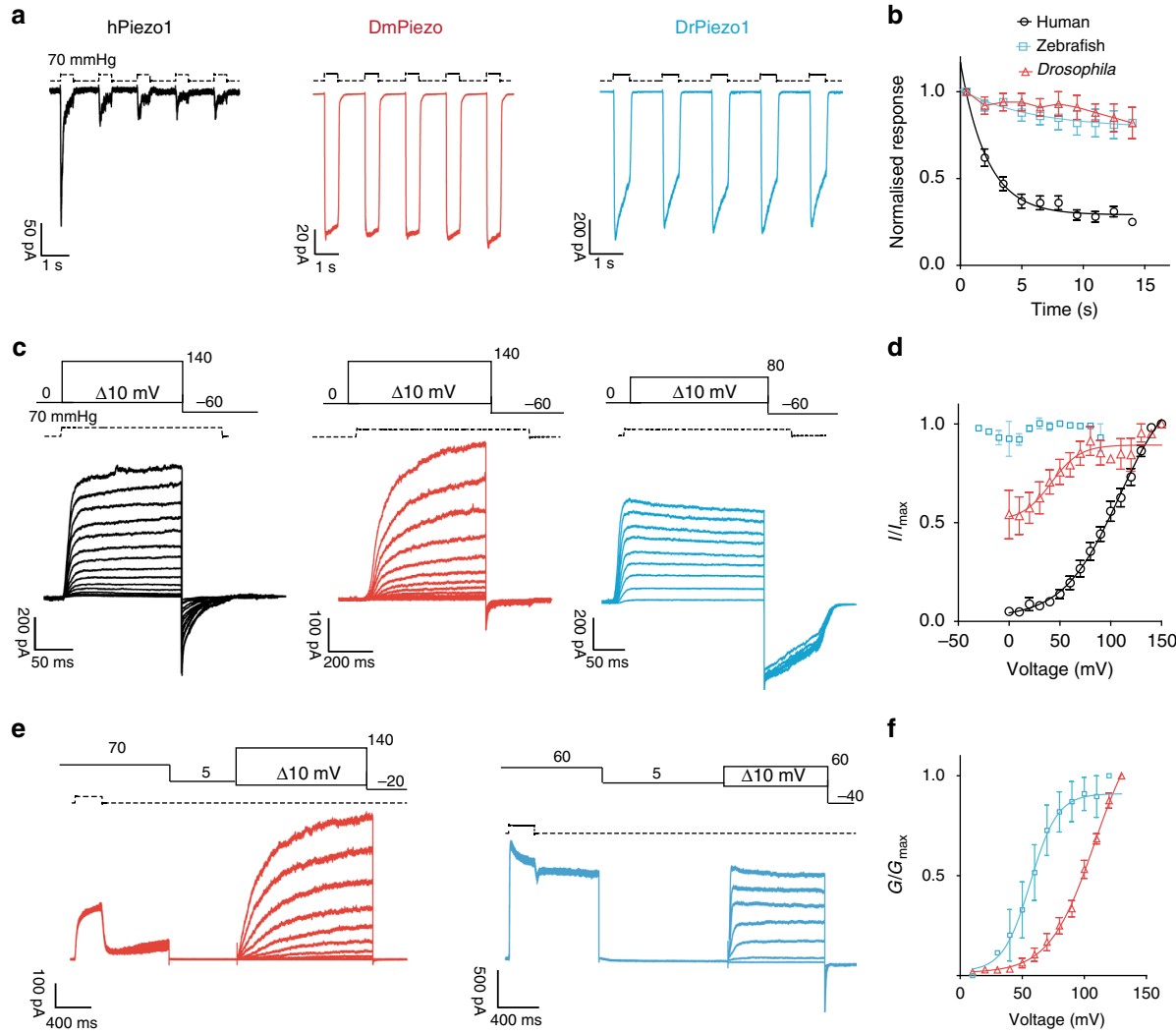

**Fig. 7** Properties of human, *Drosophila* and zebrafish PIEZO1. **a** Five pulses of saturating pressures at −60 mV were applied to patches pulled from N2a $^{Piezo1-/-}$ expressing the human (black, $n = 8$), *Drosophila* (red, $n = 5$), and zebrafish PIEZO1 (blue, $n = 9$). **b** Current amplitudes were normalized to the first-pressure pulse and plotted against time. Pooled data are shown as mean ± SEM. **c** Current responses to a pressure stimulation of 70 mmHg during 300 ms voltage steps ranging from 0 to 140 mV, followed by a repolarization step to −60 mV to obtain tail currents. **d** Tail currents from individual cells were normalized to their maximum and fitted to the Boltzmann relationship (human $V_{50} = 96.7 ± 4.8$ mV, slope 19.9 ± 0.9, 5 cells; *Drosophila* $V_{50} = 40.0 ± 9.8$ mV, slope 12.1 ± 3.9, 7 cells). Note how the pressure-mediated currents of the zebrafish PIEZO1 are insensitive to voltage. Pooled data are shown as mean ± SEM. **e** Same stimulation protocol as in Fig. 6b for DmPIEZO (red) and DrPIEZO1 (blue). **f** Macroscopic conductance values were normalized to maximum conductance and plotted against voltage to obtain a G/V curve. DmPIEZO did not reach saturation at voltages as high as 140 mV, while DrPIEZO had $V_{50}$ of 20.2 ± 6.8 mV and a slope of 7.9 ± 1.3 ($n = 7$). (See also Supplementary Figs 2, 5 and 6)

Mechanically gated hPIEZO1 currents at voltages as high as +160 mV caused the deactivation of hPIEZO1 to slow down, but no sag or reactivation phase was observed ($n = 10$, Supplementary Fig. 7). In contrast, mechanically gated currents mediated by wild-type DmPIEZO and DrPIEZO1 showed sag and reactivation kinetics (switch to voltage-gated mode) following pressure stimuli starting at 50 and 20 mV, respectively ($n = 5$, Supplementary Fig. 7). If the reactivation of the channels was indicative of a switch to voltage-gated mode, then the protocol we previously used (step to 5 mV for 1–2 s) should allow voltage gating (Fig. 6). Indeed, the opening of both DmPIEZO and DrPIEZO1 could be triggered by the application of voltage steps alone (Fig. 7e, f). Voltage gating of the wild-type DrPIEZO1 protein was robust and very similar to wild-type mPIEZO1 in the presence of Yoda1 (Fig. 6), with a $V_{50}$ of 20.2 ± 6.8 mV ($n = 7$) and a slope of 7.9 ± 1.3 with a corresponding gating charge $e_0$ of 3.8 ± 0.7. The DmPIEZO protein was also opened by positive voltage pulses, but

showed markedly slower activation kinetics and voltages as high as 140 mV still did not produce saturating activation (Fig. 7e, f). Consequently, the voltage sensitivity of DmPIEZO was shifted very significantly to the right compared to DrPIEZO1. Thus, PIEZO channels in both *Drosophila* and zebrafish can be switched to a voltage-gated mode even more readily than mammalian PIEZO proteins.

## Discussion

We show that both vertebrate and invertebrate mechanosensitive PIEZO ion channels are polymodal, being voltage modulated, and can even be directly voltage gated following a preceding mechanical stimulus. One important new feature of mammalian PIEZOs is that, under physiological resting membrane potentials, ~95% of the channels are unavailable for pressure gating. However, outward permeation of the channel driven by depolarizing

voltage rapidly removes both desensitization and the loss of inactivation that is observed with repetitive mechanical stimulation. Application of positive voltages in the presence of ionic gradients that imposes inward permeation failed to relieve PIEZO1 from desensitization. Thus, permeation rather than voltage plays a critical role in PIEZO1 gating. Indeed, with outward permeation driven by positive electrochemical gradients almost all PIEZO channels can recover from their inactivated state. The voltage-modulation and voltage gating of PIEZO channels appears to be a fundamental property of the channel pore and is dramatically altered by pathological mutations that cause xerocytosis. We have also shown that pressure stimuli given at positive potentials can transition the channel into a voltage-gated mode, a situation strongly promoted by pathogenic mutations and by a gating modulator[49]. Our data can be explained by the existence of an inactivation gate, for which a closed conformation is strongly favored at physiological resting membrane potentials. Opening of the inactivation gate is promoted by outward permeation, and in this conformation voltage gating of the channel is possible (Fig. 6h). Voltage gating is a very prominent physiological feature of older vertebrate (fish) and invertebrate (fly) PIEZO proteins. The susceptibility of PIEZO channels to voltage modulation or voltage gating are thus ancient properties of the channel that differ in prominence throughout evolution. Voltage modulation and voltage gating of PIEZO channels is likely a deep property that has been adapted to enable a variety of mechanosensing roles in many cell types and organisms.

PIEZO1 display desensitization and a loss of inactivation with repeated stimuli[5,39]. This property could be problematic for mechanosensing, where mechanical force should be constantly monitored over time. We show that pressure steps as short as 40 ms at positive voltage are sufficient to recover 90% of PIEZO channel activity, thus suggesting a critical role for the rate of flux of ions in controlling channel inactivation. Thus, slowing of inactivation and desensitization can both be reversed by outward permeation, which could happen in excitable cells under physiological conditions.

Our data could be explained by the presence of an inactivation gate, like in K2P channels, is closed by inward-ion flux and re-opened by outward permeation[36]. We also show that outward-ion permeation allows PIEZO1 to exit desensitization with slow kinetics (time course of 100 ms, at least one intermediate state) (Fig. 2e, l). Although further mutagenesis studies will be required to further confirm this hypothesis, our data strongly suggest that the ion permeation pathway plays a crucial role in determining the inactivation properties of the channel.

By using conditioning positive voltage steps, we found that it requires considerable driving force to open the inactivation gate greater than +80 mV to activate all channels (Fig. 3a, b). Nevertheless, mechanical activation of PIEZO channels combined with outward permeation shifts more channels into a non-desensitized state available to sense mechanical force. Thus voltage modulation of PIEZO channels is potentially a powerful mechanism to regulate force sensing in cells. Indeed, it has been shown that mechanosensitive currents in sensory neurons can be sensitized by inflammatory signals[51,52] and this could take place via regulation of PIEZO channels[53]. It is thus conceivable that inflammatory signals could shift voltage sensitivity of the PIEZO channels into a range where action potential firing in the presence of mechanical force could release many more channels for activation.

Despite the high amino acid sequence homology between PIEZO1 and PIEZO2, these two channels appear to be activated by different kinds of mechanical stimuli. The whole-cell patch-clamp experiments in cells have provided evidence that expression of PIEZO1 or PIEZO2 is accompanied by a fast current activated by cell indentation or substrate deflection[5,31]. PIEZO1 channel activity has also been extensively studied in excised patches and in artificial lipid bilayers where, similarly to TRAAK channels[54], membrane stretch or increased curvature are sufficient to gate the channel[20]. So far the evidence that PIEZO2 channels can be gated by membrane stretch has been lacking. Furthermore, attempts to elicit pressure-mediated currents in outside-out patches from cells overexpressing PIEZO2 failed both in the heterologous system (this report) and in Merkel cells, where PIEZO2 is abundantly expressed[6,48]. We show using PIEZO1/PIEZO2 chimeras that the N-terminal region of PIEZO1 allowed us to measure pressure activated currents in excised patches from N2a cells, in which a large portion of *Piezo1* gene had been deleted (Supplementary Fig. 3). Importantly, the kinetic properties of the mechanosensitive current were characteristic of PIEZO2 currents displaying faster inactivation kinetics than PIEZO1 (Fig. 5, Supplementary Fig. 4C). Interestingly, the pore region of the chimeric P1/P2 protein was capable of sensing voltage and outward permeation could also relieve loss of inactivation and desensitization. Indeed, we found that, at resting membrane potentials (−60 mV) the apparent open probability of the chimeric protein was as low as that found for PIEZO1 (~5–10%). Thus, the last 350 amino acids of PIEZO2 appear to code for voltage modulated pore with a putative inactivation gate that can be activated by membrane stretch when fused with the N-terminal PIEZO1 sequence. These experiments support the view that force may be transferred to the pore-forming C-terminal regions via sequences in the N-terminal region[55], but in contrast to other reports, our experiments are not complicated by the presence of PIEZO1[28]. We show that a P2/P1 chimera can be robustly activated by soma indentation with inactivation kinetics similar to PIEZO1. The P2/P1 chimera, in contrast to a P1/P2 chimera, completely loses inactivation when stimulated by pressure in an outside-out patch, which makes it hard to draw conclusions regarding the location of a voltage sensing domain in this chimeric channel. The P2/P1 chimera is primarily formed by PIEZO2, which is not responsive to direct membrane stretch per se (Fig. 5c and ref. [48]). It seems reasonable to conclude that stretch sensing domains reside within the N-terminal region of both PIEZO proteins. However during evolution, PIEZO2 might have lost the ability to transfer mechanical forces derived from membrane stretch to open the pore. By joining PIEZO2 N-terminal sequences to the pore module of PIEZO1, this lost transduction pathway appeared to be reconstituted although with altered inactivation kinetics. Alternatively, the "stretch sensor" of PIEZO1 does not reside in one region of the protein, but is dispersed across distant domains like the "blades" and the pore module[21]. Thus both modules might act together to sense changes in tension in the surrounding lipid bilayer.

Xerocytosis is a human genetic disease that has been associated with missense mutations in PIEZO1. Here we have shown that the strongest xerocytosis mutants have large effects on the voltage sensitivity of the channel, thus the R2482H/K mutation shifts the voltage modulation 60 mV leftward to dramatically increase apparent open-probability at negative potentials (Fig. 4). This is of particular relevance, in light of the fact, that erythrocytes have a membrane potential close to 0 mV[56,57]. Our data thus suggest that in erythrocytes the percentage of PIEZO1 channels available for activation could change from 5% in normal individuals to more than 40% in individuals carrying a single-point mutation (Fig. 4b).

The mutated 2482 arginine residue is located at the bottom of the IH of PIEZO1, a position close to the inner pore. Interestingly, in other ion-channels families, arginine residues at the extremities of pore-lining regions have been shown to be essential for anchoring the helices to the lipid bilayer and to stabilize the

closed conformation of the channel[58]. We thus speculate that R2482 normally stabilizes the channel in a closed conformation and movement of this residue (together with other basic ones) might confer voltage-modulation. It is worth noting in this context that arginine residues in the S1–S4 domains play a fundamental role in voltage sensing in most canonical voltage-gated ion channels[59–61]. Interestingly, voltage-modulation of pressure-gated ion channels has previously been described for the archetypical pressure sensitive bacterial channel MscS, which also requires arginine residue within the lipid bilayer[35,37,62]. However, so far there are no reports that bacterial mechanosensitive channels can be directly voltage gated.

Our data show that PIEZO1 is a polymodal channel that can be gated by voltage following a mechanical stimulus. A mutant channel involved in human disease (mouse R2482H/K) undergoes an unprecedented switch from a pressure-gated to a voltage-gated mode at voltages more positive than ~30 mV. The transition is caused by a sustained outward-ion flux in the absence of pressure. Although outside-out patches are constantly subject to a tension within the recording pipette, such tension is not sufficient to open even sensitive channels as the R2482K mutant nor PIEZO1 wt sensitized by Yoda1 (see resting current at +80 mV in Fig. 6b, e).

The switch between the two modes is characterized by a sag followed by a reactivation phase (Fig. 6). We hypothesize that switching may cause the inactivation gate to remain fully open and that PIEZO1 undergoes a conformational change that allows the channel to respond directly to voltage. Evidence for a conformational change was provided by estimating the gating charge ($e_0$), as to sense voltage, charged residues position within the electric field change upon voltage sensor activation. We estimated the gating charges from the slope of macroscopic currents to provide a lower limit of the charge movement. However, we observed a threefold increase in $e_0$ upon switching from pressure to voltage-gating mode (from 2.2 to ~6 $e_0$ for R2482K, Supplementary Table 2), suggesting that charges not available to move with electric field changes in pressure mode become free to move in voltage-gated mode because of a conformational change near the pore, a graphical model is shown in Fig. 6h. Voltage gating and voltage modulation of PIEZO channels is an ancient property present in both invertebrate (fly), older vertebrate (fish), and wild-type PIEZO channels. Comparisons of the pore regions of these channels could provide insights into the structural determinants necessary for the inactivation gate and its regulation by outward permeation. In summary, the revelation that voltage can modulate and gate mechanosensitive PIEZO channels is likely a deep property that is integral to their physiological function across a wide variety of mechanosensing cells.

## Methods

**Cell culture**. N2a cells (wt and N2a $^{Piezo1-/-}$) were cultured in DMEM/Opti-MEM (50/50) containing 5% fetal calf serum and 1% penicillin and streptomycin according to the provider's protocol (Neuro-2a ATCC CCL-131). The medium was changed to a serum-free medium before transfection. Cells were transfected with Piezo1 (a gift from Dr. Ardem Patapoutian) or chimeric Piezo1–Piezo2 cDNA constructs containing an IRES-EGFP sequence to select transfected cells. Cells were transfected in a 35 mm petri dish, pre-coated with poly-(L)-lysine, with 1 µg DNA and 3 µl of HD Fugene (Promega) following the manufacturer's protocol and were recorded 12–48 h after transfection. Only EGFP expressing cells were selected for electrophysiological recordings.

**Molecular biology**. Mutations were introduced in the mouse Piezo1 plasmid using the XL QuikChange™ Site Directed Mutagenesis (Agilent Technologies). The chimera between Piezo1 and Piezo2 was produced by PCR by joining the residues 1-2188 of mPiezo1 and 2472-2822 of mPiezo2. The resulting chimeric construct of 2539 amino acids was cloned into an IRES-EGFP containing vector by SLIC reaction (primers supplied in Supplementary Table 3).

The *D. melanogaster* Piezo was cloned from *Drosophila* mRNA (a gift from Dr. Robert Zinzen), while the *D. rerio* (zebrafish) Piezo1 was cloned from pooled individuals at 48 h post fertilization (a gift from Dr. Daniela Panakova). Both cDNA were cloned into an IRES-EGFP vector and transfected into N2a $^{Piezo1-/-}$ cells. Only EGFP expressing cells were selected for electrophysiological recordings.

**Generation of Piezo1 KO N2a cells**. Deletion of the *Piezo1* gene was generated using CRISPR/Cas9 technology. Four gRNAs targeting exon 6 and exon 45 were generated using the gRNA designer (http://crispr.mit.edu/) for the nickase Cas9 (Addgene plasmid 48140, a gift from Dr. Feng Zhang). N2a cells were transfected with gRNA and Cas9n and single cell sorted 1 week after transfection to generate a clonal *Piezo1* $^{-/-}$ cell line. The selected clone had three copies (alleles) of *Piezo1*, a deletion of at least 17 Kb in each allele. More importantly, no mechanically activated currents were detected by using soma indentation (Supplementary Fig. 2). Sequences of the gRNAs and the sequencing reactions of the resulting deleted alleles are reported in Supplementary Fig. 2.

**Electrophysiology**. Recordings were performed in excised outside-out patches pulled from N2a cells. Experiments were performed at room temperature. Recording pipettes were prepared using a DMZ puller and subsequently polished to a final resistance of 6–8 MΩ for outside-out patches. Pressure stimuli were applied through the recording pipette via a High Speed Pressure Clamp (Ala Scientific). Soma indentation experiments were performed in whole-cell configuration using microelectrodes with a resistance between 2 and 4 MΩ. Uncompensated series resistance values were less than 2 MΩ. Cells were held at −60 mV and mechanical stimulation was performed using a blunt glass probe (tip size 3–4 µm).

Experiments in outside-out configurations were performed in symmetrical ionic conditions and in a divalent-free buffer. Solutions contained (in millimolar): 140 NaCl, 10 HEPES, 5 mM EGTA adjusted to pH 7.4 with the NaOH. Whole-cell experiments were performed with the following solutions intracellularly (in millimolar): 140 KCl, ten Hepes, five EGTA, and one MgCl$_2$. Extracellular solution (in millimolar): 140 NaCl, ten Hepes, five glucose, one MgCl$_2$, and two CaCl$_2$.

For experiments where the intracellular concentrations of NaCl were altered, sucrose was used as a replacement. In Fig. 2h the following solutions were used. Intracellularly (in millimolar): ten NaCl, ten HEPES, 5 mM EGTA, 260 sucrose, pH 7.4.

Currents recorded in outside out and whole cell were sampled at 10 kHz and filtered at 3 kHz using an EPC-10 amplifier and Patchmaster software (HEKA, Elektronik GmbH, Germany). The currents were subsequently analyzed using FitMaster (HEKA, Elektronik GmbH, Germany). To calculate the single channel conductance of PIEZO1, 5–20 s long segments containing single channel openings were first filtered at 1 kHz. Subsequently, all point histograms were constructed using FitMaster. From a double Gaussian fit, the single channel conductance was calculated by plotting the average current at each potential versus the voltage. Slope conductance values at positive and negative holding potentials were obtained by plotting current–voltage relationships and by fitting the following relationship to the data points (each point was the average of 50–200 openings).

$$\gamma = I/V + C, \tag{1}$$

where $\gamma$ is the slope of the line, $I$ is the single current measured, $V$ is the voltage, and $C$ is the reversal potential (Prism 5, Graphpad). Ten to ninety percent rise times were calculated using an in-built calculation of Patchmaster. The baseline and the peak were first detected. Successively, the time that it takes for the current to go from 10% to 90% of the peak value was calculated.

Tail currents were measured at −60 mV, unless otherwise stated. Peak currents at −60 mV were leak subtracted during the analysis, normalized to the largest current and fitted to a standard Boltzmann equation:

$$f(x) = \left(\left(\text{amplitude}/1 + \exp_{1/2}^{(-(X-X)/\text{slope})}\right)\right) + \text{offset}, \tag{2}$$

where the slope corresponds to $RT/ZF$ ($R$, universal gas constant; $T$ temperature; $Z$ equivalent gating charge; $F$, Faraday constant). Individual fit from single cells were pooled together and values are expressed as mean ± SEM.

We refer to desensitization of the process that involves repetitive stimuli that lead ultimately to a decrease in channel amplitude throughout the course of the experiment. We refer to inactivation to the altered conducting state that PIEZO1 undergoes during a mechanical stimulation. This becomes apparent through the alteration of the time course of the decay of pressure-mediated currents.

**Statistics**. Parametric datasets were compared using a two-tailed, Student's t-test, paired or unpaired depending on the experimental setup. Non-parametric datasets were compared using a Mann–Whitney test. One-way ANOVA and Dunnett's post-hoc test were used as indicated in the figure legends. Data are reported as mean ± SEM.

**Data availability**. Other data are available from the corresponding authors upon reasonable request.

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

## Acknowledgements

We thank Prof. Thomas Baukrowitz for initial comments on the manuscript. Liana Kozitzki for technical assistance. This work was supported by an Alexander von Humboldt fellowship to M.M. and a Deutsche Forschungsgemeinschaft Collaborative Research Grant to G.R.L. (Project A9 SFB 958).

## Author contributions

Conceptualization: M.M. and G.R.L. Methodology: M.M. Investigation: M.M., M.R.S.-V., R.F., O.S.-C. Writing: M.M. and G.R.L. Visualization: M.M., M.R.S.-V and R.F. Supervision: M.M. and G.R.L. Project administration: M.M. Funding acquisition: G.R.L.

## Additional information

**Competing interests:** The authors declare no competing interests.

