## [Peer Review File · Nature Communications]

Reviewers' comments:

Reviewer #1 (Remarks to the Author):

In this manuscript Moroni et al. present a previously unidentified role of transmembrane voltage in the gating of mechanically activated Piezo ion channels.

They show that Piezo1 open probability is voltage dependent, and that positive voltage and/or outward permeation remove inactivation. They continue showing that mutations related to human disease affect this process, and data from a chimera suggest that this process is influenced by the C-terminal part of the protein. Finally, they investigate the variation and conservation of this mechanism by comparing the function of several orthologues.

The manuscript is interesting, because the authors develop several creative and well thought out protocols to gain a significant amount of insight into what is obviously a very complicated mechanism in Piezo channels. The results are significant, because this unusual mode of gating might have wide physiological implications, which are beyond the scope of the biophysical exploration presented here, but are well-addressed in the discussion.

Although I am overall enthusiastic about this work, the manuscript has a major weakness in that quite many conclusions are not fully justified by the data and in that several interpretations are far overreaching. Some of these points can be taken care of by editing the text alone, but in some cases more experiments are absolutely required for the conclusions to hold up:

Points that require experiments:

1. The conclusion that the recovery from the desensitized state depends on outward permeation and not voltage needs more support. The reasons are that conditions for establishing the direction of permeation have not been worked out clearly enough, and that the conclusion is very important for the interpretation of several pieces of data that follow. To begin with, the reversal potential is estimated with a voltage-ramp protocol (Figure 2G), which is not ideal for an inactivating channel. As it stands right now, the minimal amount of current for P4 in Figure 2H is not much different from the result shown in 2F where there is not current elicited. A statistical quantification of this inward current is also not provided. Therefore, experiments shown in Figure 2H should be repeated, but a stronger chemical gradient should be used to demonstrate that outward permeation at a negative voltage is still sufficient for recovery. Additionally, a stronger chemical gradient should be used to generate larger inward permeation to show that inward permeation at a positive voltage is insufficient for recovery.

2. A change in the kinetics of recovery from inactivation would be a potential mechanistic explanation for the data, as opposed to the opening of an inactivation gate. For example, DmPiezo and zPiezo1 (Figure 7A) could simply have a faster recovery from inactivation than hPiezo1, and/or its voltage dependence could be different; also data in Figure 2L could be explained by this. Therefore, this possibility needs to be explored experimentally. In both cases (Figure 2L and 7A) the authors should repeat experiments, but vary the time delay between pulses, and then the amplitude ratio should be plotted against this time interval. Next, testing a range of voltages for the paired pulse protocol is also necessary to probe for a potential voltage dependency of recovery.

3. The claim that the pore is responsible for the voltage gating mechanism needs more support. The chimeric construct presented does not rule out the possibility that the voltage gating mechanism is not part of the Piezo1 portion, including the anchor which has been implicated in ion selectivity. This is important as the vast majority of the chimeric channel is still Piezo1. A reverse chimera should be generated to test this more carefully, or this alternate possibility should be prominently discussed.

Points that require editing:

4. The authors conclude that Piezos are “directly voltage-gated” and I find this conclusion wrong, as none of the data show that voltage alone, i.e. without a preconditioning pressure stimulus can activate Piezo. This mechanism is therefore better described as “indirectly voltage-gated”. The authors should change and discuss this clearly.

5. The claim that the transition between the inactive and active state occurs proportional to the “amount of ions” that flows through the channel is premature. The authors should show this either through experiments with greater chemical gradients (point 1) or analysis of transition dependency on length of conditioning pre-pulse. If not, this conclusion has to be changed.

6. The claim that the last two transmembrane domains mediate inactivation is also overreaching, as the swapped domain also include the very large C-terminal extracellular domain and the C-terminal tail. Without evidence from a reverse chimera (point 3), claims that the pore module definitively mediates voltage gating should be downplayed in interpretation.

7. Experiments in Figure 6B are conducted under the premise that the channel is “without a mechanical stimulus”. However, there are no data that support this notion and it is most certainly is inaccurate, as there is always resting tension within any patch configuration. The conclusion that Piezo1 is thus gated “without a mechanical stimulus” (line 624) needs to be removed.

8. In Figure 3D and Figure 4B, the wild-type current does not saturate, which fundamentally flaws the measurement of a shift in $V_{1/2}$. → The resulting conclusions do therefore not hold up and should be removed from the manuscript.

Minor points to improve the manuscript:

9. The distinction between desensitization and inactivation should be clearly defined at the beginning of the manuscript. While it seems likely that the two processes are mechanistically connected, it should be made clear throughout the manuscript which process each experiment addresses.

10. CRISPR N2A cells were only introduced in the middle of the study, which is okay. Negative control experiments similar to Figure 1A and Figure 2E showing no endogenous currents would rule out any ambiguity in experiments using non CRISPR lines, and would be nice to add.

11. The structural resolution outside of the CED and the pore module is not high, which might not be clear to the non-expert reader. Therefore, when making reference to the structure (e.g. exact locations of point mutations) this fact should be mentioned.

12. Diagrams in Figures 3 and 6 are not easy to interpret. Each part of the image should be labeled with its gating mode and state (ie. ‘Voltage gated’-‘closed’, ‘Pressure gated’-‘inactivated’, etc.), and the precise stimulus between all panels (above rate arrows) should be clearly defined. For example, it is not clear in Figure 3E, why configuration 3 is more favorable than config. 2. In Figure 3E it is also unclear why outward permeation drives config. 4 to config. 3. In Figure 6H, it is unclear why in pressure-gated mode, outward permeation promotes both an opening and closing of the inactivation gate (bottom left config.), or why inward permeation in the pressure-gated mode drives the channel towards an open voltage-gated config.

13. There are mistakes in panel lettering in Figure 2. J and K are skipped in the figure and legends for panel I onwards are mislabeled.

Reviewer #2 (Remarks to the Author):

In this manuscript, Moroni and colleagues present compelling data documenting an original finding of the voltage modulation of Piezo1 channels on desensitization. They find that desensitization of the channel with repeated stimulation can be overcome by outward flux of ions and suggest that this is caused by an inactivation gate. They investigated various human mutations of Piezo1 and show that the voltage modulation is present to varying degrees in the mutant channels and also have varying percentages of active channels at physiological voltages. Additionally, they find that Piezo1 can be pushed into a state where it is directly voltage regulated when there are certain pore modifications with mutations, but also when it is modulated by Yodh1. They further show that voltage modulation and direct voltage gating are ancient properties of the channel by looking at drosophila and zebrafish Piezo1. Additionally their data suggests the pressure mediated membrane stretch sensitivity is attributed to the N-term part of Piezo1. Overall, the manuscript presents interesting findings about Piezo1 function through proposing an inactivation gate that can be opened with voltage as well as a direct voltage gate on the channel that can be unmasked with modulation of the channel. This work will open up multiple lines of investigation into how the inactivation mechanism can be modulated and what physiological mechanisms are involved to shift the voltage modulation and increase the number of available channels.

Major Comments:

1. Is there any visual evidence that for the direct voltage gating no longer has any membrane stretch? Some videos of the pressure stimulation can be done on the patch and look at changes in the curvature of the patch.
2. Fig 2L and M: How were the rise times calculated? The light blue trace appears to have multiple components to the rise, where was the actual peak for the light blue considered? The timing for the major part of the rise does not look so different between the 2 conditions, but maybe a zoom on the onset would help. It appears that the light blue has the added slow increase which could be the process of the removal of inactivation. If the time dependence of the release of inactivation was investigated, this would be clearer. Understanding this will also indicate whether excitable cells may be able to release the block when their time at highly depolarized potentials is often short.
3. Fig 3 and Line 255: Is it the rate of flux that removes inactivation or total flux. If total flux is the determining factor then a longer time at lower potentials should also continually increase the amount of release from inactivation. Again, investigating the time dependence would be helpful.
4. The experiments with the chimera Piezo1-Piezo2 can be the result of the N-term of Piezo1 providing the voltage regulation and allosterically affecting the pore region. Why not perform a similar set of experiments on the native full length Piezo2 using probe stimulation? These major observations should not be largely affected by space clamp or clamp speed issues which the excised patch excels with. In fact, Lewis et al. (2017 Cell Rep 19(12):2572–2585) recently showed that inactivation of mouse Piezo2 can be modulated by an N terminal mutation I802F, suggesting it may be the N-term portions that are responsible.
5. Figure 7C: the ratio of current at +140mV as compared to the repolarization current amplitude at -60mV look much larger for DPiezo than the other Piezos. Why is this the case? Is it that the stimulus was not saturating? Or is the inactivation of DmPiezo almost instantaneous at negative potentials?

Minor Comments:

Line 78: None of the mutations completely remove voltage modulation, it is just decreased sensitivity to voltage modulation.

Line 116: Should this be the maximum apparent open probability, since it refers to the maximum

number of channels that are not inactivated?

Line 177: Reference should be 2I, not 2G.

Line 183: The 0.25 value is not the value seen in Fig 2I.

Line 188: In the legend, you need to correct the reference to panel letters and order of discussion.

Line 289: Doesn't removing inactivation mean increasing the apparent open probability? Aren't these two things the same?

Line 571: Should be DmPIEZO

It would be better to show the data points with the mean and std deviation shown, rather than bars.

Reviewer #3 (Remarks to the Author):

The authors present an intriguing study in which they demonstrate that mechanosensitive PIEZO channels are voltage modulated using excised outside-out patch clamp electrophysiology with tail current protocols. They also demonstrate that mutations in Piezo1 underlying the human disease Xerocytosis confer the ability of Piezo1 to be directly gated by voltage. Wildtype Zebrafish Piezo1 is shown to be voltage gated as well, leading the authors to suggest that voltage gating and modulation are ancient properties of the Piezo channels.

Interestingly, the authors also provide evidence that mammalian PIEZOs exhibit extremely low open channel probability at resting membrane potentials. Repeated pressure stimulation leads to desensitization and loss of inactivation, but outward permeation via depolarization restores sensitivity and inactivation, presumably by opening an inactivation gate in the pore region. Yoda1, a small molecule shown to modulate Piezo1 activity, was also able to promote transitions to a voltage-gated mode similar to the Xerocytosis mutants.

To examine voltage modulation in Piezo2, they created a chimera of Piezo1 and Piezo2, since Piezo2 is not activated by their stretch protocol. Attaching the N-terminal of Piezo1 to the C-terminal region in Piezo2 (containing the pore region) allowed them to test the stretch activated responses of Piezo2. Examination of the tail currents in cells expressing the chimera revealed that Piezo2 is also subject to voltage modulation.

Overall, this is an interesting study. If validated by other labs, this work will have a great impact on Piezo channel biophysics and mechanosensation as a whole. I am therefore happy to support its publication.

Specific points:

1. Figure 3e, PIEZO states 2 and 3 should be light green, not states 3 and 4.
2. They made an N2a cell line lacking Piezo1 to exogenously express their chimeric protein without the confound of background Piezo1 expression. However, it is unclear whether N2a cell lines express Piezo2. Could endogenous Piezo2 be altering the results of their chimera analysis?
3. The identity of the inactivation gate remains a mystery after this study. There is little discussion of

this in the manuscript, except to say that it shares features with K2P channels. Given that the presumed inactivation gate is what binds all the elements of the present study, this is surprising. Is there structural or sequence homology between the K2P inactivation gate and the Piezos?

4. It isn't clear why N2a cells were used for this study. It would strengthen the study to demonstrate that some of the main findings are observable in other cell types. For instance, does expression of the R2482 Piezo1 mutants confer voltage gating in a second cell line?

5. Line 955, it isn't clear what "stretch of openings were first filtered at 1KHz" means.

6. Figure S4 A and B are not sufficiently labeled

Reply to reviewers

Reviewers' comments:

Reviewer #1 (Remarks to the Author):

In this manuscript Moroni et al. present a previously unidentified role of transmembrane voltage in the gating of mechanically activated Piezo ion channels.

They show that Piezo1 open probability is voltage dependent, and that positive voltage and/or outward permeation remove inactivation. They continue showing that mutations related to human disease affect this process, and data from a chimera suggest that this process is influenced by the C-terminal part of the protein. Finally, they investigate the variation and conservation of this mechanism by comparing the function of several orthologues.

The manuscript is interesting, because the authors develop several creative and well thought out protocols to gain a significant amount of insight into what is obviously a very complicated mechanism in Piezo channels. The results are significant, because this unusual mode of gating might have wide physiological implications, which are beyond the scope of the biophysical exploration presented here, but are well-addressed in the discussion.

Although I am overall enthusiastic about this work, the manuscript has a major weakness in that quite many conclusions are not fully justified by the data and in that several interpretations are far overreaching. Some of these points can be taken care of by editing the text alone, but in some cases more experiments are absolutely required for the conclusions to hold up:

Response: We are delighted that the reviewer appreciates the significance and novelty of our data. We have addressed all the reviewers concerns either with a substantial amount of new experimental data but also with clarifications to the text which are detailed below.

Points that require experiments:

1. The conclusion that the recovery from the desensitized state depends on outward permeation and not voltage needs more support. The reasons are that conditions for establishing the direction of permeation have not been worked out clearly enough, and that the conclusion is very important for the interpretation of several pieces of data that follow. To begin with, the reversal potential is estimated with a voltage-ramp protocol (Figure 2G), which is not ideal for an inactivating channel. As it stands right now, the minimal amount of current for P4 in Figure 2H is not much different from the result shown in 2F where there is not current elicited. A statistical quantification of this inward current is also not provided. Therefore, experiments shown in Figure 2H should be repeated, but a stronger chemical gradient should be used to demonstrate that outward permeation at a negative voltage is still sufficient for recovery. Additionally, a stronger chemical gradient should be used to generate larger inward permeation to show that inward permeation at a positive voltage is insufficient for recovery.

To begin with, the reversal potential is estimated with a voltage-ramp protocol (Figure 2G), which is not ideal for an inactivating channel.

Response: The reviewer is right in stating that a ramp protocol might be inappropriate to calculate the E_{rev} for fast inactivating channels. However, the ramp protocol we used was from positive to negative

potentials and Piezo1 does not inactivate during the time course required to reach its reversal. Thus the Erev calculation for Piezo1 from positive the predicted Erev was not subject to inactivation biases. However, to confirm the measured reversal potentials we also performed a new standard I/V determination using voltage steps and, comparing the two methods side by side, we obtained almost identical Erev values.

Changes: see revised Figure S1 and text line 220-223.

As it stands right now, the minimal amount of current for P4 in Figure 2H is not much different from the result shown in 2F where there is not current elicited. A statistical quantification of this inward current is also not provided

Response: The reviewer probably did not see that the scale bar between Fig 2F and 2G has a 10-fold difference. This means that the inward current passing through PIEZO1 at 40mV (P4 in Fig 2G) is significantly larger than 0. We have now made this clear by adding a bar graph (Fig 2I) with the proper statistical analysis as suggested by the reviewer.

Changes: line 223-224, 207-208.

Therefore, experiments shown in Figure 2H should be repeated, but a stronger chemical gradient should be used to demonstrate that outward permeation at a negative voltage is still sufficient for recovery. Additionally, a stronger chemical gradient should be used to generate larger inward permeation to show that inward permeation at a positive voltage is insufficient for recovery.

Response: We have tried the experiments suggested by the reviewer. Thus, we tried to record currents in presence of 2mM and 5mM NaCl in the intracellular pipette and adjusting the osmolarity with Sucrose. However, such low concentrations of ions do not allow stable recordings due to the lack of intracellular ions to stabilize the contact between the membrane patch and the pipette. We have therefore not succeeded in performing these experiments. We have carried out further experiments by using Choline-Cl and NMDG-Cl. However, in absence of cations Cl ions show considerable permeability through PIEZO1, thus shifting the voltage to values lower than 40mV, as previously reported (Coste et al. 2010)

2. A change in the kinetics of recovery from inactivation would be a potential mechanistic explanation for the data, as opposed to the opening of an inactivation gate. For example, DmPiezo and zPiezo1 (Figure 7A) could simply have a faster recovery from inactivation than hPiezo1, and/or its voltage dependence could be different; also data in Figure 2L could be explained by this. Therefore, this possibility needs to be explored experimentally. In both cases (Figure 2L and 7A) the authors should repeat experiments, but vary the time delay between pulses, and then the amplitude ratio should be plotted against this time interval. Next, testing a range of voltages for the paired pulse protocol is also necessary to probe for a potential voltage dependency of recovery.

Response: The reviewer suggests a mechanism that has previously been excluded by Gottlieb et al (2012). In this report it was shown that PIEZO1 enters a permanent non-inactivating state that seems to be independent of the time that is allowed for PIEZO1 to recover. Our investigation confirms these findings. To reiterate this point we have repeated the paired pulses experiments in Fig 2J at 2 sec and 30 sec intervals. Furthermore, as suggested by the reviewer, we have performed a detailed analysis (Fig S1 C and D) of the time and the voltage that is required for PIEZO1 to recover from its inactive state. Our findings show that brief positive pulses at positive potentials as short as 50ms allow PIEZO1

to exit inactivation.

Changes: Figure S1 C and D. Figure 2K and L, line 236-247.

3. The claim that the pore is responsible for the voltage gating mechanism needs more support. The chimeric construct presented does not rule out the possibility that the voltage gating mechanism is not part of the Piezo1 portion, including the anchor which has been implicated in ion selectivity. This is important as the vast majority of the chimeric channel is still Piezo1. A reverse chimera should be generated to test this more carefully, or this alternate possibility should be prominently discussed.

Response: We have gladly taken the reviewer's suggestion and constructed a reverse PIEZO2/PIEZO1 chimera. We have analysed the properties of this new chimera in detail and the results have been added to the MS (Figure 5). The P2/P1 chimera was functional and expressed in a manner comparable to PIEZO1 in soma indentation. However, findings in outside-out patches were confounded by the absence of any inactivation kinetic for this construct. We reasoned that the P2/P1 chimeric construct was not capable of gating pressure stimuli as efficiently as PIEZO1 or P1/P2 chimera (current levels were 10 fold less than PIEZO1 and or P1/P2 chimera).

Changes: Figure 5 and corresponding figure legend, lines 383-397, 407-464.

Points that require editing:

4. The authors conclude that Piezos are "directly voltage-gated" and I find this conclusion wrong, as none of the data show that voltage alone, i.e. without a preconditioning pressure stimulus can activate Piezo. This mechanism is therefore better described as "indirectly voltage-gated". The authors should change and discuss this clearly.

Response: We agree completely with the point made. It is a very unusual type of voltage gating that we have discovered as it requires a mechanical pre-pulse. We have now changed the text throughout the MS to make this clear. Changes: line 499, 530-532.

5. The claim that the transition between the inactive and active state occurs proportional to the "amount of ions" that flows through the channel is premature. The authors should show this either through experiments with greater chemical gradients (point 1) or analysis of transition dependency on length of conditioning pre-pulse. If not, this conclusion has to be changed.

Response: we have now carried out new experiments that directly address this point. The new data are shown in Figure S1 and using a paired pulse protocol we could show that the transition of states depends on both the duration of the mechanical pre-pulse and the value of the preceding voltage step. In Figure S1C and D we now show that pulses as short as 50ms at +60mV are sufficient to recover 90% of PIEZO1 channels from the inactive state. A maximal recovery is achieved with pulses at 80mV.

Changes: Figure S1, line 236-247.

6. The claim that the last two transmembrane domains mediate inactivation is also overreaching, as the swapped domain also include the very large C-terminal extracellular domain and the C-terminal tail.

Without evidence from a reverse chimera (point 3), claims that the pore module definitively mediates voltage gating should be downplayed in interpretation.

Response: the reviewer makes a very valid point and indeed our new data with the reverse P2/P1 chimera shows that the situation is more complicated than we initially supposed. We have changed the wording in the text accordingly.

Changes: line 396, 410, 413,444,461,724,717-729

7. Experiments in Figure 6B are conducted under the premise that the channel is “without a mechanical stimulus”. However, there are no data that support this notion and it is most certainly is inaccurate, as there is always resting tension within any patch configuration. The conclusion that Piezo1 is thus gated “without a mechanical stimulus” (line 624) needs to be removed.

Response: This is a valid point we have reworded the text accordingly.

Changes: line 497, 528-529

8. In Figure 3D and Figure 4B, the wild-type current does not saturate, which fundamentally flaws the measurement of a shift in $V_{1/2}$.– The resulting conclusions do therefore not hold up and should be removed from the manuscript.

We are aware that it the current is not fully saturating. We have observed a high variation in saturation levels depending on the patch. While some patches saturated at 140mV others did not. The fundamental problem is that it is probably that the currents saturate at voltages that are above the technical limitations (with higher voltages we simply lose the recording). In Fig 6G we recorded PIEZO1 wt with the Yoda1 and showed that under these circumstances currents saturate already at potentials as low as 50mV. If the black curve for wt PIEZO1 (Fig 3D and 4B) saturates at even higher voltages the calculated V_{50} is clearly an underestimation of the reported shift. This is a technical limitation of the experiment that does not fundamentally alter the fact that we can shift the voltage dependence. We have added a comment in the results section that draws the reader’s attention to this problem.

Changes: line 118-121

Minor points to improve the manuscript:

9. The distinction between desensitization and inactivation should be clearly defined at the beginning of the manuscript. While it seems likely that the two processes are mechanistically connected, it should be made clear throughout the manuscript which process each experiment addresses.

Response: desensitization and inactivation are clearly two tightly connected events in channel kinetic. We specifically refer to desensitization in experiments that involve successive stimulations of the channel that lead to a decrease in current amplitude. We refer to inactivation to the altered conducting state that PIEZO1 undergoes during a mechanical stimulation. This becomes apparent through the alteration of the time course of the decay of pressure-mediated currents. We have now clarified the two concepts in the methods section. Changes line: 1022-1025

10. CRISPR N2A cells were only introduced in the middle of the study, which is okay. Negative control

experiments similar to Figure 1A and Figure 2E showing no endogenous currents would rule out any ambiguity in experiments using non CRISPR lines, and would be nice to add.

Response: N2a do express endogenous mouse PIEZO1 currents at a very low level (Figure S3 C and Coste et al. 2010). This made it difficult to make reliable recordings from endogenous PIEZO1 in outside out patches taken from N2A cells, although it was sometimes possible. We regularly used the PIEZO1 knockout cell line to express mutant channels as well as Drosophila and Zebrafish orthologues. In these later experiments we regularly over expressed wild type channels on a Piezo1 knockout background and obtained identical results to those experiments done with PIEZO1 channels overexpressed in wild type N2A cells. Thus, we have no reason to believe that the use of these two different cells is a confounding factor in our experiments.

11. The structural resolution outside of the CED and the pore module is not high, which might not be clear to the non-expert reader. Therefore, when making reference to the structure (e.g. exact locations of point mutations) this fact should be mentioned.

Response: During the submission of the manuscript two new structures of PIEZO1 have been published, enhancing the resolution of previous ones and confirming the position of the residues within each domain (Guo and MacKinnon, 2017; Saotome et al., 2017). Although the new structures are about 3.6 Å and do not give provide atomic resolution of side chains, the Xerocytosis mutations are now well mapped and have been confirmed to reside in specific domains. The mutation R1353P, which was initially thought to reside on a peripheral helix, has now been resolved to reside on the so called “beam” domain. The manuscript has been changed accordingly and we have added references to the new available structures.

Changes: lines 330-331, 340-341

12. Diagrams in Figures 3 and 6 are not easy to interpret. Each part of the image should be labeled with its gating mode and state (ie. ‘Voltage gated’-‘closed’, ‘Pressure gated’-‘inactivated’, etc.), and the precise stimulus between all panels (above rate arrows) should be clearly defined. For example, it is not clear in Figure 3E, why configuration 3 is more favorable than config. 2. In Figure 3E it is also unclear why outward permeation drives config. 4 to config. 3. In Figure 6H, it is unclear why in pressure-gated mode, outward permeation promotes both an opening and closing of the inactivation gate (bottom left config.), or why inward permeation in the pressure-gated mode drives the channel towards an open voltage-gated config.

Response: We have modified the figure legends to explain the model in more detail as suggested by the referee. The equilibrium arrow were erroneously depicted in the figure and they have now been changed. **Changes:** Figure 3 E, 6 H and figure corresponding figure legends.

13. There are mistakes in panel lettering in Figure 2. J and K are skipped in the figure and legends for panel I onwards are mislabeled.

Response: Thank you for pointing this out, the errors have been corrected in the revised MS. **Changes:** lines 207-216.

Reviewer #2 (Remarks to the Author):

In this manuscript, Moroni and colleagues present compelling data documenting an original finding of the voltage modulation of Piezo1 channels on desensitization. They find that desensitization of the channel with repeated stimulation can be overcome by outward flux of ions and suggest that this is caused by an inactivation gate. They investigated various human mutations of Piezo1 and show that the voltage modulation is present to varying degrees in the mutant channels and also have varying percentages of active channels at physiological voltages. Additionally, they find that Piezo1 can be pushed into a state where it is directly voltage regulated when there are certain pore modifications with mutations, but also when it is modulated by Yodh1. They further show that voltage modulation and direct voltage gating are ancient properties of the channel by looking at drosophila and zebrafish Piezo1. Additionally their data suggests the pressure mediated membrane stretch sensitivity is attributed to the N-term part of Piezo1. Overall, the manuscript presents interesting findings about Piezo1 function through proposing an inactivation gate that can be opened with voltage as well as a direct voltage gate on the channel that can be unmasked with modulation of the channel. This work will open up multiple lines of investigation into how the inactivation mechanism can be modulated and what physiological mechanisms are involved to shift the voltage modulation and increase the number of available channels.

Response: We appreciate that the reviewer finds our manuscript of interest. We have addressed the reviewers remaining concerns below.

Major Comments:

1. Is there any visual evidence that for the direct voltage gating no longer has any membrane stretch? Some videos of the pressure stimulation can be done on the patch and look at changes in the curvature of the patch.

Response: We are unfortunately unable to perform simultaneous visual and electrophysiological recordings with our experimental set up. Previous attempts to record membrane curvature in the patch of an outside-out patch were also hard to interpret due to the high cytosolic content in the pipette after patch excision ((Lewis and Grandl, 2015)).

We believe that hardly any pressure remains in the patch and if a minimal amount of pressure remains, it is insufficient to elicit opening of PIEZO1 per se. Figure 6A and B show the extremely sensitive mutant R2482K. The P50 of this mutant is much lower than PIEZO1 wt (Albuisson et al., 2013; Bae et al., 2013). In these panels patches are stepped first to a range of voltages, followed by pressure. As no channel opening is observed before applying external pressure it is plausible to assume that the amount of “background” pressure present between the patch and pipette is not sufficient to elicit channel opening. Our data cannot rule out the possibility that a minimal tension within the dome of the patch persists, however this tension is not sufficient to trigger channel opening and would not confound our results.

Where appropriate we have pointed out in the text that no additional membrane stretch is applied to the patch during the application of voltage stimuli. Changes: line 499, 530-531

2. Fig 2L and M: How were the rise times calculated? The light blue trace appears to have multiple components to the rise, where was the actual peak for the light blue considered? The timing for the major

part of the rise does not look so different between the 2 conditions, but maybe a zoom on the onset would help. It appears that the light blue has the added slow increase which could be the process of the removal of inactivation. If the time dependence of the release of inactivation was investigated, this would be clearer. Understanding this will also indicate whether excitable cells may be able to release the block when their time at highly depolarized potentials is often short.

The rise time was calculated with 10-90% procedure built into the Patchmaster software. Essentially the peak and the baseline are calculated in the area defined by cursors and the time between the 10% of the peak and the 90% of the peak was calculated. Due to the complex shape of the rise phase, most likely due to the slow removal of inactivation, no fitting was possible. A more complete analysis, including a multi state model, of this process is required to find a fitting function capable of describing the mechanism. Changes: Methods 1013-1014

If the time dependence of the release of inactivation was investigated, this would be clearer. Understanding this will also indicate whether excitable cells may be able to release the block when their time at highly depolarized potentials is often short.

Response: Very good point and we have now performed new experiments to address this issue. We have added the new data to the manuscript see Figure 2 K, L and Figure S1. The new data in Figure S1 describe that using a paired pulse protocol the transition between states depends on both the duration of the mechanical pre-pulse and the value of the preceding voltage step. In Figure S1C and D we now show that pulses as short as 50ms at +60mV are sufficient to recover 90% of PIEZO1 channels from the inactive state. A maximal recovery is achieved with pulses at 80mV. Changes: Figure S1, lines 236-247

3. Fig 3 and Line 255: Is it the rate of flux that removes inactivation or total flux. If total flux is the determining factor then a longer time at lower potentials should also continually increase the amount of release from inactivation. Again, investigating the time dependence would be helpful.

Response: This is a good point which we have addressed with new experimental data see revised MS Figure 2K, L and Figure S1. In figure S1 we show that the rate of flux rather than the total flux seems to be more determining for the release PIEZO1 from inactivation. A 500ms pulse at 60mV releases from inactivation as many channels (~90%) as a pulse of a 40ms at the same voltage.

Changes: Figure S1, 238-247, 671-672

4. The experiments with the chimera Piezo1-Piezo2 can be the result of the N-term of Piezo1 providing the voltage regulation and allosterically affecting the pore region. Why not perform a similar set of experiments on the native full length Piezo2 using probe stimulation? These major observations should not be largely affected by space clamp or clamp speed issues which the excised patch excels with. In fact, Lewis et al. (2017 Cell Rep 19(12):2572–2585) recently showed that inactivation of mouse Piezo2 can be modulated by an N terminal mutation I802F, suggesting it may be the N-term portions that are responsible.

Response: Probe or poking stimulation does not lead to a mechanical stimulus that equally reaches every part of the cell in which PIEZO1 resides. We find this method to be too variable to make meaningful comparisons. We have, however added new experiments with a reverse chimera which also allows us to make conclusions about the possible location of the voltage sensor (ie probably

distributed, see answer to reviewer 1 above). Changes: Figure 5 and corresponding figure legend, lines 383-397, 407-464.

5. Figure 7C: the ratio of current at +140mV as compared to the repolarization current amplitude at -60mV look much larger for DPiezo than the other Piezos. Why is this the case? Is it that the stimulus was not saturating? Or is the inactivation of DmPiezo almost instantaneous at negative potentials?

Response: We found that the inactivation on repolarization is incredibly fast for DmPIEZO. This is something that we want to investigate further and that may give us insights on inactivation properties of the PIEZO channel family.

Minor Comments:

Line 78: None of the mutations completely remove voltage modulation, it is just decreased sensitivity to voltage modulation.

Response: Agreed we have changed the text accordingly. Changes: lines 79, 355

Line 116: Should this be the maximum apparent open probability, since it refers to the maximum number of channels that are not inactivated?

Response: The reviewer is correct, thanks for point this out. We have corrected this mistake. Line 116.

Line 177: Reference should be 2I, not 2G.

Response: We have corrected this mistake

Line 183: The 0.25 value is not the value seen in Fig 2I.

Response: We have corrected this mistake. The trace was not representative and has been changed (now Figure 2 F).

Line 188: In the legend, you need to correct the reference to panel letters and order of discussion.

Response: This has been corrected

Line 289: Doesn't removing inactivation mean increasing the apparent open probability? Aren't these two things the same?

Response: Yes, the sentence was misleading and it has been corrected. Changes: line 294-295

Line 571: Should be DmPIEZO

Response: This has been corrected.Changes: line 592

It would be better to show the data points with the mean and std deviation shown, rather than bars.

Response: We believe that error bars express the content of very dense figures such as the ones in this manuscript in a more communicative way than individual data points. We will follow the editorial

guidelines once the manuscript is accepted.

Reviewer #3 (Remarks to the Author):

The authors present an intriguing study in which they demonstrate that mechanosensitive PIEZO channels are voltage modulated using excised outside-out patch clamp electrophysiology with tail current protocols. They also demonstrate that mutations in Piezo1 underlying the human disease Xerocytosis confer the ability of Piezo1 to be directly gated by voltage. Wildtype Zebrafish Piezo1 is shown to be voltage gated as well, leading the authors to suggest that voltage gating and modulation are ancient properties of the Piezo channels.

Interestingly, the authors also provide evidence that mammalian PIEZOs exhibit extremely low open channel probability at resting membrane potentials. Repeated pressure stimulation leads to desensitization and loss of inactivation, but outward permeation via depolarization restores sensitivity and inactivation, presumably by opening an inactivation gate in the pore region. Yoda1, a small molecule shown to modulate Piezo1 activity, was also able to promote transitions to a voltage-gated mode similar to the Xerocytosis mutants.

To examine voltage modulation in Piezo2, they created a chimera of Piezo1 and Piezo2, since Piezo2 is not activated by their stretch protocol. Attaching the N-terminal of Piezo1 to the C-terminal region in Piezo2 (containing the pore region) allowed them to test the stretch activated responses of Piezo2. Examination of the tail currents in cells expressing the chimera revealed that Piezo2 is also subject to voltage modulation.

Overall, this is an interesting study. If validated by other labs, this work will have a great impact on Piezo channel biophysics and mechanosensation as a whole. I am therefore happy to support its publication.

Response: We are delighted that the reviewer has found our study of interest we have answered all open questions with new experimental data and corrections to the text.

Specific points:

1. Figure 3e, PIEZO states 2 and 3 should be light green, not states 3 and 4.

Response: We have changed the figure accordingly.

2. They made an N2a cell line lacking Piezo1 to exogenously express their chimeric protein without the confound of background Piezo1 expression. However, it is unclear whether N2a cell lines express Piezo2. Could endogenous Piezo2 be altering the results of their chimera analysis?

Response: This is a good point of course. However, no expression data for Piezo2 was found in the RNAseq analysis of the study "A Global Regulatory Mechanism for Activating an Exon Network

Required for Neurogenesis” (Raj et al., 2014). Additionally, we did not find basal/endogenous activity of MA ion channels in the N2a^{Piezo1^{-/-}} cells when assayed with pressure clamp and indentation (Figure S3 A).

3. The identity of the inactivation gate remains a mystery after this study. There is little discussion of this in the manuscript, except to say that it shares features with K2P channels. Given that the presumed inactivation gate is what binds all the elements of the present study, this is surprising. Is there structural or sequence homology between the K2P inactivation gate and the Piezos?

Response: Before our study it was no real evidence for an inactivation gate that could be controlled by voltage. Since there is no structural homology between K2P and PIEZO channels an extensive mutagenesis study would be needed to identify the molecular identity of the inactivation gate.

4. It isn't clear why N2a cells were used for this study. It would strengthen the study to demonstrate that some of the main findings are observable in other cell types. For instance, does expression of the R2482 Piezo1 mutants confer voltage gating in a second cell line?

Response: Since most of the experiments were done with the mouse PIEZO1 clone we wanted to use a cell line with a mouse background, to make sure that all components necessary for mechanotransduction would not be influenced by differences between orthologues.

Most of the experiments were initially done in HEK cells and the recorded data in these cells showed no difference in comparison to the data recorded in N2a (data not shown). However, patches recorded from HEK cells were not able to withstand voltages >100mV with concomitant application of >70mmHg of pressure. When we switched to using N2a cells we found that patches could be exposed to mechanical stretch and high positive voltages without losing the recording.

5. Line 955, it isn't clear what 'stretch of openings were first filtered at 1KHz' means.

Response: We realized that the term “stretch” in the context of MA ion channels might cause confusion. Stretch was referring to the portion of the recording sweep. We have changed the wording. Changes: line 1001-1002

6. Figure S4 A and B are not sufficiently labeled

Response: The required additional labels have been expanded and corrected

REVIEWERS' COMMENTS:

Reviewer #1 (Remarks to the Author):

The authors have done a great job in addressing all my concerns and suggestions for improvement. This is now a high-quality manuscript and of highest interest.

Reviewer #2 (Remarks to the Author):

In this revision, Moroni et al addressed my major points sufficiently, however, some editing and clarifications are required. With these edits, I fully support the manuscripts publication.

Line 59: The peripheral regions of the protein are composed of the extracellular "blade" domains, 24 peripheral helices (PHs) in each subunit and intracellular "beam" and "anchor" domains (Saotome et al., 2017).

24 peripheral helices are resolved, but as Guo and MacKinnon point out, there are likely 3 other "Piezo" repeats that likely also exist and were unresolvable.

Line 188 and Fig 2H: the ratio of P5/P1 was decreased to 0.25 ± 0.5 (n=13) (Figure 2F and H). The value of 0.25 ± 0.5 is still not the value shown in Figure 2H. The green bar in Figure 2H does not have an average value of 0.25 nor does it have an error bar of 0.5.

Line 211: The pulses ratio P5/P1 for E (white) ,F (green) and G (orange) are shown and are statistically significant, (Anova P= 0.00004 Dunnett's post-hoc test, n=10 , dF = 18)

The n = 10 value is different from the value listed in the text on line 188, which says n=13. You actually would need to specify the n for both the orange and the green. Also, shouldn't there be 2 sets of statistical parameters for the comparison between white and green, and white and orange?

Line 226 and elsewhere:

V should be capitalized in mV.

Line 230: Furthermore, the ratio between P5/P1 was 0.28 ± 0.6 (n=7) (Fig 2 H).

The value listed for the orange ration in Fig 2H does not look to have the right error bar, since it does not show a 0.6 error. Also the error bar for the orange is smaller than for the green bar which is reported to have an error of 0.5.

Fig S1 C & D

Please state when the normalized amplitude is taken. I would take the amplitude near the beginning of the 3rd pressure step in order to assay the % of current recovered by the 2nd pulse (test pulse that is varied). If using toward the end of the 3rd pulse, then there is more recovery that can occur during the long 1 second pulse that is not solely indicative of the change caused by the short or higher voltage steps from the 2nd pressure pulse.

???Line 297: The conditioning pressure pulse at positive potentials removes ~80% of the inactivation (Figure 3C)

Where is the 80% value taken from? I would think that you are comparing the values in 3D where you show the I-V curve with and without the conditioning step, where I would expect at some potential, there is a delta I/Imax of 0.8 based on the statement, but this does not exist, the max delta I see is around 0.3. Alternatively, you can look at the inactivation at positive potentials in Fig 3C, where it would be helpful to highlight the 100mV trace for the 2nd pulse to see how you arrived at 80%.

Line 420-421: remove "be"

Line 505: should be "completely"

Line 510-512: Strikingly, both the R2482H and R2482K mutant channels were activated by the voltage steps in a manner similar to classical voltage-gated ion channels with a V_{50} of 50.7 ± 9.3 (10 cells R2482H).

The V_{50} shown on the graph in 6G does not look like a value of 50.7 mV. The value for the blue line at a $G/G_{max} = 0.5$ is closer to 35 mV

Line 577-578: an estimation of the gating charges from the steepness of the Boltzmann (Figure 6 G) fit yielded a value of $5.2 e_0$.

The Boltzmann curve in 6G for Yodh1 with mPiezo1 is steeper than the either of the mutations R2482K or R2482H, however, you are reporting a lower e_0 . This does not make sense, since a steeper slope should results in a higher e_0 . The pressure gated curve does look shallower than the voltage gated mode curve, so these values make sense that the e_0 for pressure gating mode is lower than that for voltage gated mode.

Looking at the methods (line 1034), the equation for the Boltzmann is incorrect. It is missing an open parenthesis and should be:

$f(x) = (\text{amplitude} / (1 + \exp(-(X - X_{1/2}) / \text{slope}))) + \text{offset}$

Additionally, the slope term used in the equation is often referred to as slope factor since it is inversely related to the slope. Changing this terminology will make it easier to understand that a smaller reported value means a steeper slope.

REVIEWERS' COMMENTS:

Reviewer #1 (Remarks to the Author):

The authors have done a great job in addressing all my concerns and suggestions for improvement. This is now a high-quality manuscript and of highest interest.

Reviewer #2 (Remarks to the Author):

In this revision, Moroni et al addressed my major points sufficiently, however, some editing and clarifications are required. With these edits, I fully support the manuscripts publication.

Line 59: The peripheral regions of the protein are composed of the extracellular “blade” domains, 24 peripheral helices (PHs) in each subunit and intracellular “beam” and “anchor” domains (Saotome et al., 2017).

24 peripheral helices are resolved, but as Guo and MacKinnon point out, there are likely 3 other “Piezo” repeats that likely also exist and were unresolvable.

We appreciate the necessary adjustments that the referees suggested and we are happy to provide clarifications. Line 59 (Revised MS Line 52) has been modified to include the unresolved but predicted peripheral helices.

Line 188 and Fig 2H: the ratio of P5/P1 was decreased to 0.25 ± 0.5 (n=13) (Figure 2F and H). The value of 0.25 ± 0.5 is still not the value shown in Figure 2H. The green bar in Figure 2H does not have an average value of 0.25 nor does it have an error bar of 0.5.

Thank you for pointing this out. The bar graphs have been corrected to match the reported values in the text. Within the error bar value we had mistakenly omitted a 0. The actual error corresponds to 0.05, not 0.5.

Line 211: The pulses ratio P5/P1 for E (white) ,F (green) and G (orange) are shown and are statistically significant, (Anova $P=0.00004$ Dunnett’s post-hoc test, n=10 , dF = 18)
The n = 10 value is different from the value listed in the text on line 188, which says n=13. You actually would need to specify the n for both the orange and the green. Also, shouldn’t there be 2 sets of statistical parameters for the comparison between white and green, and white and orange?

The n number has been corrected appropriately for all data sets with the corresponding values for statistical significance.

Line 226 and elsewhere:
V should be capitalized in mV.

This has been corrected

Line 230: Furthermore, the ratio between P5/P1 was 0.28 ± 0.6 (n=7) (Fig 2 H).

The value listed for the orange ration in Fig 2H does not look to have the right error bar, since it does not show a 0.6 error. Also the error bar for the orange is smaller than for the green bar which is reported to have an error of 0.5.

The bar graphs have been corrected to match the reported values in the text. Within the error bar value we had omitted a 0. The actual error corresponds to 0.06

Fig S1 C & D

Please state when the normalized amplitude is taken. I would take the amplitude near the beginning of the 3rd pressure step in order to assay the % of current recovered by the 2nd pulse (test pulse that is varied). If using toward the end of the 3rd pulse, then there is more recovery that can occur during the long 1 second pulse that is not solely indicative of the change caused by the short or higher voltage steps from the 2nd pressure pulse.

We completely agree with the reviewer's consideration. The amplitude values for normalization were initially taken at the end of the 3rd pulse. We have reanalyzed the data and replotted the normalized value. The plotted graphs refer to the averaged amplitude values between 10 and 20 ms after the application of the third pressure step. The values were chosen as 5 to 10 ms which is the approximate time that is required for the pressure clamp to achieve steady state pressure.

Line 297: The conditioning pressure pulse at positive potentials removes ~80% of the inactivation (Figure 3C)

Where is the 80% value taken from? I would think that you are comparing the values in 3D where you show the I-V curve with and without the conditioning step, where I would expect at some potential, there is a $\Delta I/I_{max}$ of 0.8 based on the statement, but this does not exist, the max ΔI I see is around 0.3. Alternatively, you can look at the inactivation at positive potentials in Fig 3C, where it would be helpful to highlight the 100mV trace for the 2nd pulse to see how you arrived at 80%.

We refer to figure 3A and B and not figure 3C. We apologize for the mistake, it has now been corrected.

Line 420-421: remove "be"

This has been corrected

Line 505: should be "completely"

This has been corrected

Line 510-512: Strikingly, both the R2482H and R2482K mutant channels were activated by the voltage steps in a manner similar to classical voltage-gated ion channels with a V_{50} of 50.7 ± 9.3 (10 cells R2482H).

The V50 shown on the graph in 6G does not look like a value of 50.7 mV. The value for the blue line at a $G/G_{max} = 0.5$ is closer to 35 mV

The reported values are correct, however, we have made a mistake when transferring the graphs into Adobe Illustrator. We have now imported the graphs correctly and they match with the reported data. The figure has been corrected accordingly

Line 577-578: an estimation of the gating charges from the steepness of the Boltzmann (Figure 6 G) fit yielded a value of 5.2 e_0 .

The Boltzmann curve in 6G for Yodh1 with mPiezo1 is steeper than the either of the mutations R2482K or R2482H, however, you are reporting a lower e_0 . This does not make sense, since a steeper slope should result in a higher e_0 . The pressure gated curve does look shallower than the voltage gated mode curve, so these values make sense that the e_0 for pressure gating mode is lower than that for voltage gated mode.

Looking at the methods (line 1034), the equation for the Boltzmann is incorrect. It is missing an open parenthesis and should be:

$$f(x) = (\text{amplitude} / (1 + \exp(-(X - X_{1/2}) / \text{slope}))) + \text{offset}$$

Additionally, the slope term used in the equation is often referred to as slope factor since it is inversely related to the slope. Changing this terminology will make it easier to understand that a smaller reported value means a steeper slope.

The gating charge estimation had a high degree of variation. Although the slope of graph in fig 6G might seem steeper for Piezo1 wt (Yoda), the slope values are not significantly different as reported in Supplementary Table 2.

In addition, we have added a parenthesis to the Boltzmann equation. We have exchanged the term “slope factor” for “slope”